# In vivo clonal expansion and phenotypes of hypocretin-specific CD4+ T cells in narcolepsy patients and controls

Wei Jiang [1,2]*, James R. Birtley [3,9,11], Shu-Chen Hung[1,2,11], Weiqi Wang [4,11], Shin-Heng Chiou[5], Claudia Macaubas[1,2], Birgitte Kornum[6], Lu Tian[7], Huang Huang [5], Lital Adler[1,10], Grant Weaver [3], Liying Lu[3], Alexandra Ilstad-Minnihan[1], Sriram Somasundaram [1], Sashi Ayyangar[1], Mark M. Davis [2,5,8], Lawrence J. Stern [3] & Elizabeth D. Mellins [1,2]*

Individuals with narcolepsy suffer from abnormal sleep patterns due to loss of neurons that uniquely supply hypocretin (HCRT). Previous studies found associations of narcolepsy with the human leukocyte antigen (HLA)-DQ6 allele and T-cell receptor α (TRA) J24 gene segment and also suggested that in vitro-stimulated T cells can target HCRT. Here, we present evidence of in vivo expansion of DQ6-HCRT tetramer+/TRAJ24+/CD4+ T cells in DQ6+ individuals with and without narcolepsy. We identify related TRAJ24+ TCRαβ clonotypes encoded by identical α/β gene regions from two patients and two controls. TRAJ24-G allele+ clonotypes only expand in the two patients, whereas a TRAJ24-C allele+ clonotype expands in a control. A representative tetramer+/G-allele+ TCR shows signaling reactivity to the epitope $HCRT_{87-97}$. Clonally expanded G-allele+ T cells exhibit an unconventional effector phenotype. Our analysis of in vivo expansion of HCRT-reactive TRAJ24+ cells opens an avenue for further investigation of the autoimmune contribution to narcolepsy development.

[1] Department of Pediatrics–Human Gene Therapy, Stanford University School of medicine, Stanford, CA 94305, USA. [2] Stanford Immunology, Stanford University School of Medicine, Stanford, CA 94305, USA. [3] Department of Pathology, University of Massachusetts Medical School, Worcester, MA 01655, USA. [4] Human Immune Monitoring Center, Stanford University School of Medicine, Stanford, CA 94305, USA. [5] Institute for Immunity, Transplantation and Infection, Department of Microbiology and Immunology, Stanford University School of Medicine, Stanford, CA 94305, USA. [6] Danish Center for Sleep Medicine, Department of Clinical Neurophysiology, Rigshospitalet, 2600 Glostrup, Denmark. [7] Department of Biomedical Data Science, Stanford University School of Medicine, Stanford, CA 94305, USA. [8] Howard Hughes Medical Institute, Stanford University, Stanford, CA 94305, USA. [9] Present address: UCB Pharma, Slough SL13WE, UK. [10] Present address: Department of Biological Regulation, Weizmann Institute of Science, Rehovot 7610001, Israel. [11] These authors contributed equally: James R. Birtley, Shu-Chen Hung, Weiqi Wang. *email: wjiang6@stanford.edu; mellins@stanford.edu

With a prevalence of 25–50/100,000 people[1], type 1 narcolepsy (T1N) with cataplexy is a sleep disorder that currently lacks a cure. It results from the loss of HCRT-producing neurons in the hypothalamus, causing an undetectable level of HCRT in the cerebrospinal fluid[2]. An autoimmune etiology for T1N has been proposed for decades, based on the discovery of an HLA class II association[3]. The vast majority (~98%) of narcoleptic patients carry DQB1*06:02 (in association with DQA1*01:02 encoding DQ6α/β heterodimers), compared to ~25% in normal individuals[4]. In addition, genome-wide association studies (GWAS) in DQ6+ individuals revealed a risk (odds ratio~1.7) conferred by a single-nucleotide polymorphism (SNP) haplotype, rs1154155-rs1483979. The latter position, with G/C alleles, is located in the TRAJ24 gene region[5–7]. However, how the two alternative alleles correlate with T1N from a functional perspective remains elusive. Several other GWAS-identified risk SNPs are located within genes that encode proteins functioning in T cell survival and class II antigen presentation, including the purinergic receptor P2YR11, cathepsin H, and the co-stimulatory molecule OX40 ligand[6,8,9]. As CD4+ T helper ($T_h$) orchestrate Ab and cytolytic T lymphocyte (CTL) responses, possible contributions to T1N from autoantibodies[10–14] and CD8+ T cells[15,16], complement the genetic data in implicating a role for CD4+ T cells in the immunopathophysiology of T1N.

A fundamental question that limits our understanding of the proposed autoimmune origin of T1N is how to determine the disease-specific feature of a self-reactive CD4+ T cell, given that the peripheral T cell repertoires in both patients and healthy individuals include diverse self-reactive clones[17]. Indeed, several studies observed comparable in vitro proliferative responses of CD4+ T cells isolated from T1N and DQ6 allele-matched healthy donors[18–20]. Recently, one study reported elevated responses of some patient-derived CD4+ T cells targeting HCRT, although most responding cells were DR-restricted[21], suggesting that it is challenging to use in vitro bulk analysis to pinpoint disease-associated clones[7]. In order to focus on DQ6-restricted clones, another study used DQ6-HCRT$_{peptide}$ tetramers to assess in vitro stimulated CD4+ T cells at both bulk and single-cell levels[7]. However, tetramer positivity alone has been demonstrated to be insufficient for the estimation and representation of functional reactivity of T cell clones[22].

To overcome difficulties arising from in vitro assays, we investigate in vivo TCR clonotypic signatures and related phenotypic characteristics at the single-cell level. We determine the structural homology between HCRT-derived peptides that bind DQ6 and identify CD4+ T cells that express TRAJ24 and bind the corresponding DQ6-HCRT$_{peptide}$ tetramers. Importantly, we discover features of tetramer+/TRAJ24+ cells in some T1N patients that differ from related clones found in some DQ6+ controls. These include: (1) the pairing of the TRAJ24+ TCRα chain with specific β chains, both chains using public complementarity-determining region 3 (CDR3) sequences, (2) the ability of such TCRαβ heterodimer to transduce functional signals in response to DQ6 presentation of the C-terminal end of a physiologically processed HCRT neurotransmitter, (3) the in vivo expansion of cells bearing these TCRαβ clonotypes with the specific TRAJ24-G allele, and (4) the expression by the expanded cells of transcriptional markers indicating an unconventional in vivo T effector ($T_{eff}$) phenotype with cytolytic potential. Our identifications of epitopes in HCRT and in vivo expanded TRAJ24-G allele+ HCRT-reactive TCR clonotypes advance the current understanding of HCRT-related autoimmunity and suggest future directions for narcolepsy research.

## Results

**Rationale for approach.** HCRT is the only protein known to be unique to neurons lost in T1N[2] and can be targeted by in vitro-stimulated T cells[21]. Therefore, we focused on identification of DQ6-binding HCRT peptides and used DQ6-HCRT$_{peptide}$ tetramers to investigate ex vivo CD4+ T cells (Fig. 1). As self-reactive or tetramer+ clones may exist in all DQ6+ individuals[7,18–20], in vitro reactivity or tetramer binding of polyclonal T cells, especially when analyzed in bulk[7,21], is unlikely to reveal autoimmune features of specific TCR clonotypes. We therefore directly sequenced ex vivo DQ6-HCRT$_{peptide}$ tetramer+/CD4+ single cells from DQ6+ individuals with/without T1N symptoms using a well-established pipeline[23,24]. Our strategy (Fig. 1) provides direct information about in vivo clone size, an essential feature associated with physiologic immune responses, and also provides direct information about in vivo phenotype[25–27]. We examined TCRαβ sequences to identify in vivo clonal expansions, defined by identical variable (V), diversity (D), and joining (J) gene regions and shared CDR3 sequences. Among these, we focused on those that express TRAJ24, given their potential disease-relevance. Previous work shows that HLA-peptide tetramer binding does not ensure TCR functional signaling in pathogen-specific T cells[22,24,28] or tumor-infiltrating CTLs[29,30]. In light of this, single-cell tetramer sorting assay likely also overestimates the truly reactive T cell population in an autoimmune context. Thus, our strategy to identify disease-associated features in epitopes and in T cells does not simply rely on bulk comparison of polyclonal tetramer+ T cells between patients and controls; instead, the staining by tetramers served as a prerequisite for isolation of self-reactive candidates for single-cell analysis. A rigorous validation for TCR function was then performed in a TCR-deficient Jurkat-reporter system[24] to identify truly reactive clonotypes that can signal in response to DQ6-HCRT$_{peptide}$.

**Epitopes and binding registers in HCRT.** To identify DQ6-restricted T epitopes in prepro-HCRT (the HCRT precursor), we used a modified version of the peptide-loading assay[31] in which HCRT peptides were tested for the ability to inhibit DQ6-binding of EBV$_{486–500}$, a known DQ6-binding epitope derived from Epstein-Bar virus[32]. We evaluated overlapping 15-mer peptides, offset by four amino acids (aa), covering the entire prepro-HCRT sequence (Supplementary Data 1). Nine peptides showed moderate to strong competitive binding (53–97.4% inhibition of the indicator EBV peptide) to DQ6 (Fig. 2a). These were consistent with in silico peptide-binding predictions using the NetMHCIIpan3.2 software[33] (Supplementary Fig. 1a–c). The nine peptides span five regions (i–v) of prepro-HCRT. A strong binder (>75% competition), HCRT$_{1–15}$, was within the signal peptide region (i) and contained a 9-aa core, LPSTKVSWA, previously shown to bind DQ6 by X-ray structure[34]. Two overlapping peptides span the C-terminus of the signal peptide and N-terminus of the secreted HCRT1 neurotransmitter. This region (ii) contained three possible registers as predicted by NetMHCIIpan3.2 using nonamers (Supplementary Fig. 1d). Using 15-mers, the algorithm only predicted two strong cores (Supplementary Fig. 1c), and our empirical-binding data further argued that the 9-aa core SSGAAAQPL present in the strong binder, HCRT$_{25–39}$, is the dominant register (Fig. 2a). Prepro-HCRT is processed intracellularly to two neurotransmitters, HCRT1 and HCRT2 known to interact with the HCRT receptors, HCRTR1 and HCRTR2, with different affinities[35]. Interestingly, five of the DQ6-binding peptides were from highly homologous regions (iii) and (iv) at the C-terminal of each of the two processed neurotransmitters. Each region contains a NetMHCIIpan-predicted

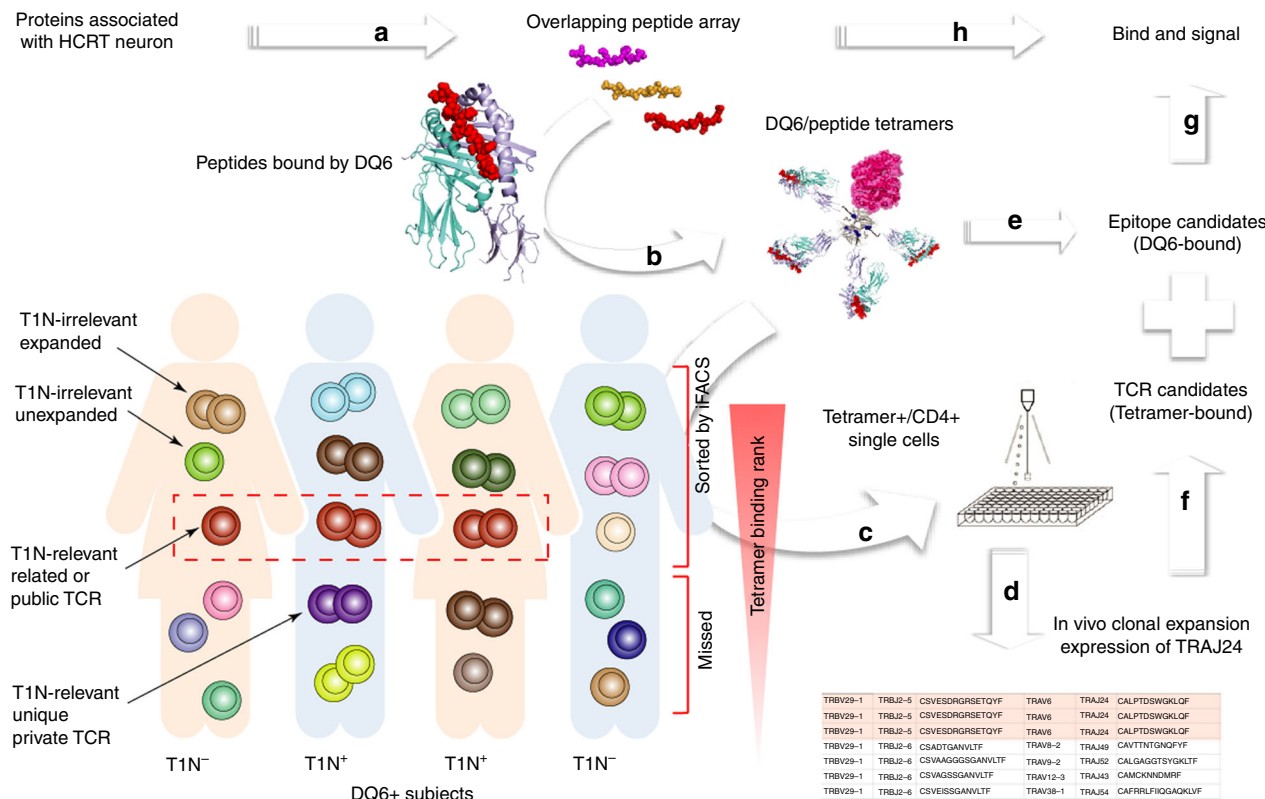

**Fig. 1** Schematic illustration of approach. **a** An array of overlapping peptides covering the entire candidate autoantigen HCRT is tested for HLA-DQ6 binding. **b** DQ6-HCRT$_{peptide}$ tetramers are synthesized based on the DQ6-binding cores determined in **a**. **c** Ex vivo CD4$^+$ T cells are isolated from PBMCs of DQ6$^+$ patients and controls using negative selection by magnetic-activated cell sorting (MACS); purified CD4$^+$ T cells are co-stained with candidate tetramers and antibodies distinguishing cell types (i.e., anti-CD4, anti-CD19). DQ6-HCRT$_{peptide}$ tetramer$^+$/CD4$^+$ cells are sorted by single cell index sorting (iFACS). Given the discordance between tetramer positivity and true autoreactivity, the iFACS-sorted cells likely include tetramer$^+$ clones with various tetramer-binding ranks, clone size, and disease-relevance: T1N-relevant (darker clones expressing public/related TCRs or unique/private TCRs) or irrelevant (lighter clones); expansion is indicated using doublets. Clones expressing TCR risk gene alleles may also be observed in DQ6$^+$ controls, as DQ6-restricted selection may occur similarly in patients and controls and T1N development is thought to rely largely on antigen-driven clonal expansion. **d** Deep sequencing of TCR and phenotypic transcripts in sorted single cells allows further assessment of T1N-associated gene signatures of tetramer$^+$ clones including both in vivo clonal expansion and expression of the TCR risk gene. **e–h** The DQ6-restricted TCR sequences are validated for ability to generate an expressed α/β TCR that binds relevant tetramers and signals after stimulation with relevant peptide epitopes. The first author created this figure

register (NHAAGILTL or NHAAGILTM), also implicated by empirical binding data (Fig. 2a). The ninth peptide was a weak binder (<75% competition) located at the prepro-HCRT C-terminal region (v), which is removed during processing to generate functional HCRT2. No strong binding register was predicted for this peptide (Supplementary Fig. 1c, d).

We further investigated cores of the strong-binding peptides using X-ray crystallography. The structure of DQ6-HCRT$_{56–69}$ bound to DQ6 (Supplementary Table 1) confirmed the predicted 9-aa core register (NHAAGILTL, Fig. 2b). Comparing the DQ6-HCRT$_{56–69}$ structure with the previously determined DQ6-HCRT$_{1–13}$ structure[34], we observed several conformational changes in α-helices of the DQ6α/β dimer (Fig. 2c and Supplementary Fig. 1e). However, the conformation of most of the DQ6 framework that would face a TCR was unchanged. We built on the structure of DQ6-HCRT$_{56–69}$ to model HCRT$_{25–37}$ and HCRT$_{87–100}$ binding to DQ6 (Fig. 2d, e), and to infer candidate TCR-facing residues. Unlike the predicted TCR contact position P5K in HCRT$_{1–13}$, neither P5G in HCRT$_{56–69}$ nor P5G in HCRT$_{87–100}$ provided a side chain that could contribute to engagement with TCR, suggesting that TCR recognition of these complexes relied on P2/P3, or P8. HCRT$_{25–37}$ contained a DQ6-binding core with relatively short side chains at all TCR-facing residues: P2S, P3G, P5A, and P8P. Together, these findings

predict that DQ6-restricted TCRs might bind to multiple HCRT epitopes.

**Various DQ6-HCRT tetramers stain CD4$^+$ T cells**. We next constructed four DQ6 tetramers using peptides with strong binding cores and tested their ability to stain CD4$^+$ T cells isolated from peripheral blood mononuclear cells (PBMCs) of T1N donors. Tetramer-staining of T cells harboring HCRT-binding capability only rarely showed a discrete positive population (Fig. 3), unlike that seen with cells recognizing pathogen-derived epitopes such as EBV$_{486–500}$ (Supplementary Fig. 2). The observed tetramer$^+$/CD4$^+$ T cell frequency of 0.039 ± 0.0029% was consistent with a low frequency of circulating class II tetramer-positive cells, as previously described[26,36–38]. Like DQ6-EBV$_{486–500}$ tetramer$^+$ cells, which could be enriched in vitro (Supplementary Fig. 2), frequencies of DQ6-HCRT$_{56–69}$ or DQ6-HCRT$_{87–100}$ tetramer$^+$ cells from T1N donors were significantly increased after in vitro stimulation with the corresponding peptides (Fig. 3a, b), confirming the existence of circulating HCRT tetramer$^+$ cells and the feasibility of using DQ6-HCRT$_{peptide}$ tetramers to isolate cells for further analysis. Cells enriched in vitro showed specificity for the peptide stimulator, as staining with HCRT$_{1–13}$ tetramer did not appreciably increase (Fig. 3a, b). However, we detected in vitro enrichment for DQ6-HCRT$_{56–69}$

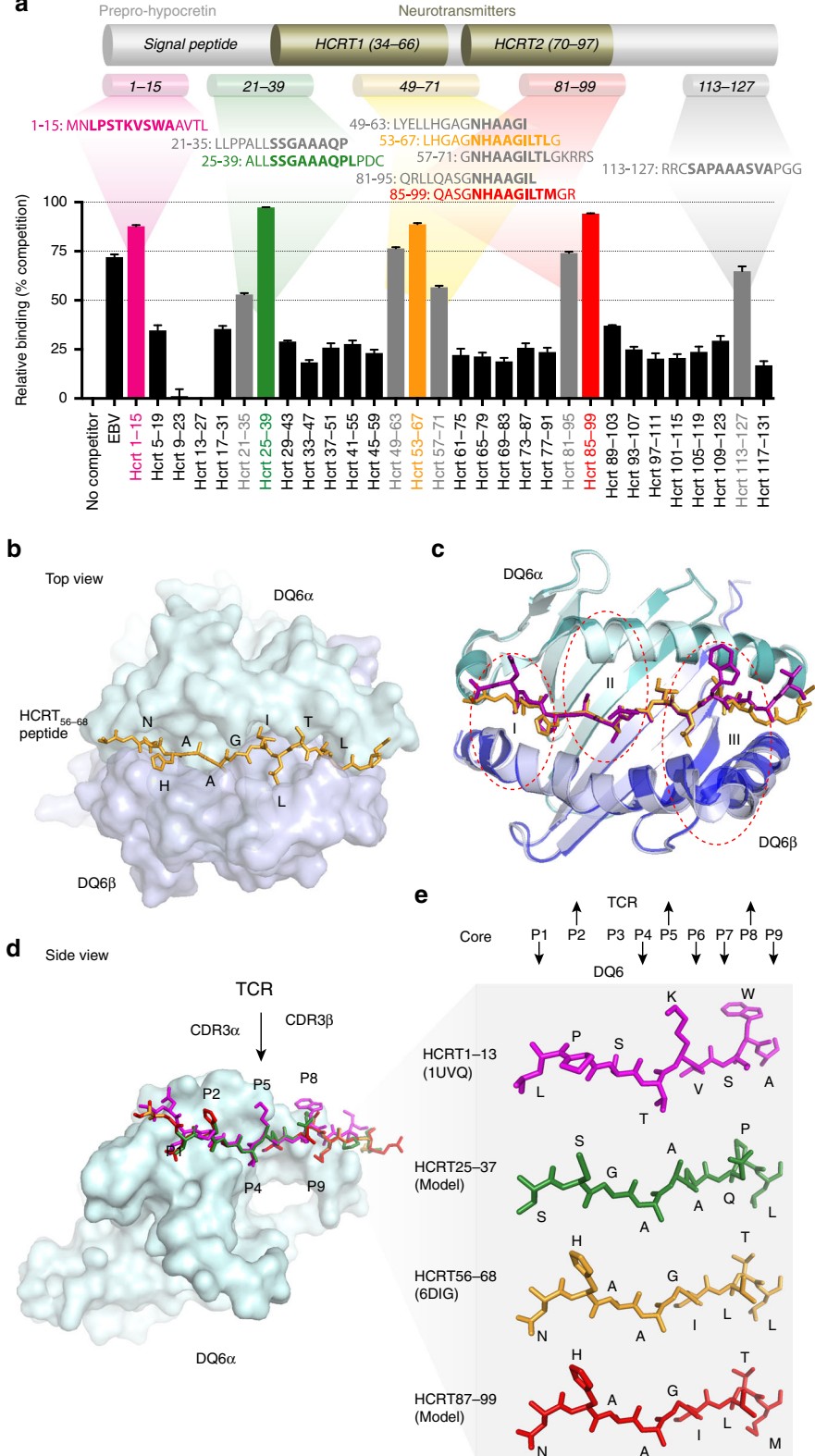

tetramer[+] cells in HCRT[87–100]-stimulated cells from some T1N donors (Fig. 3c). This likely reflects the high homology between HCRT[58–66] and HCRT[89–97] registers and also confirms the structural prediction that some DQ6-restricted TCRs bind multiple epitopes from HCRT (Fig. 2e).

To avoid alteration of in vivo clone size by in vitro stimulation, we used DQ6-HCRT[peptide] tetramers to directly stain and analyze

ex vivo CD4[+] T cells from DQ6[+] donors. These tetramers each detected similar frequencies of circulating CD4[+] T cells (including expanded and unexpanded clones) from patients and controls (Fig. 3d). We then sorted comparable numbers of DQ6-HCRT[peptide] tetramer[+]/CD4[+] T cells from patients and controls using single cell index sorting (iFACS)[39] for further comparisons of in vivo expanded clones. Prior to sequencing analysis, we used

**Fig. 2** DQ6-binding cores in prepro-HCRT and their structural impacts. **a** Thirty overlapping peptides covering prepro-HCRT were added, individually, into a reaction containing soluble DQ6, bio-EBV$_{486-500}$ peptide, and HLA-DM (a peptide loading catalyst). DQ6-associated bio-EBV$_{486-500}$ was measured at steady state by ELISA[31]. %Competition = 1−%DQ6/EBV binding. Strong (>75%Competition, in colors) and weak (50–75%Competition, in gray) DQ6 binders with predicted cores (bolded) are aligned. Data are represented as mean ± SEM (standard error of the mean); $n = 4$. **b** Top view of HCRT$_{56-69}$ (orange) in the peptide-binding groove of DQ6 (α/β, light green/blue surface, PDB: 6DIG). Core residues are indicated. **c** Alignment of DQ6-HCRT$_{56-69}$ (α/β, light green/blue cartoon; peptide, orange stick) and DQ6-HCRT$_{1-13}$ structures[34] (α/β, dark green/blue; peptide, magenta) illustrating three regions in DQ6 with noticeable conformation differences (detailed in Supplementary Fig. 1e). **d** Side view of DQ6 (β chain removed to reveal peptides) in complex with HCRT$_{1-13}$ (PDB: 1UVQ) and DQ6-HCRT$_{56-69}$ (PDB: 6DIG) and models of HCRT$_{25-37}$ and HCRT$_{87-100}$ (sticks in the same color as in **a**). Arrow indicates predicted positioning for interaction of TCRα/β CDR3s. **e** Zoom-in of the 9-aa core registers of peptides shown in **d**. Arrows indicate TCR (up) or DQ6 (down) facing resides

mean index fluorescence intensity (FI, recorded by iFACS) of tetramer binding signal to rank tetramer$^+$ cells. We found that circulating DQ6-HCRT$_{peptide}$ tetramer$^+$ cells showed comparable ranks on average between patients and controls (Fig. 3e and Supplementary Fig. 3). These initial comparisons provided the basis to further compare frequencies of tetramer$^+$ clones that had been expanded in vivo (see below), which likely reflects antigen-driven responses.

**Case/control cells with different tetramer specificities**. We next analyzed the sequencing output. iFACS screening of 3–5 million CD4$^+$ T cells from each donor was necessary for the isolation of one 96-well plate of single cells (Supplementary Fig. 4). We used blindly paired case/control PBMC samples for each experiment (Table 1) to reduce effects of technical variations on downstream analyses of case/control differences. In total, 5503 wells of sorted individual cells were analyzed using the established algorithm[23] (Supplementary Data 2). TCR transcripts were detected in 4605 wells (83.7% well coverage). As commonly observed with this approach[23,24,36], not every well yields called TCRαβ from raw sequencing reads. Out of 2762 sequenced cells that had paired TCRαβ and productive α/β CDR3 sequences, 1492 cells were from 30 case plates and 1270 cells were from 28 control plates (Table 1 and Supplementary Data 3), indicating the absence of technical bias towards either cohort. However, unique to DQ6-HCRT$_{87-100}$ tetramer$^+$/CD4$^+$ cells, there were significantly more called TCRαβ from cases than from controls (Supplementary Fig. 5a). This suggests a biological difference that survives technical variations inherent in independent sort/sequence experiments of case/control pairs. One possibility is that some patient cells have generated more copies of TCR RNA due to activation at the stage of in vivo clonal expansion, which then favors successful single-cell sequencing.

As in vivo clonal expansion of T cells has been identified as a feature of clones related to autoimmune diseases[40], we analyzed expanded clonotypes to further compare case/control samples with different HCRT specificities. Overall, 52 TCRαβ clonotypes (~1% of all sorted cells) had multiple isolates from the same donor (Supplementary Data 3). Similar to the overall lowest frequency of DQ6-HCRT$_{25-37}$ tetramer$^+$ cells (Fig. 3d), DQ6-HCRT$_{25-37}$ tetramers detected significantly fewer expanded clonotypes (1 from a control C7 out of 443 cells with called TCRαβ sequences) compared to the other tetramers (Supplementary Fig. 5b). These results are consistent with HCRT$_{25-37}$ peptide rarely being presented in vivo, likely due to it spanning a border between the signal peptide and HCRT1 and harboring a site for proteolytic cleavage. Unlike the other three DQ6-HCRT$_{peptide}$ tetramer specificities that yielded a slightly skewed detection of more expanded clonotypes in control samples, DQ6-HCRT$_{87-100}$ tetramers identified expanded clonotypes in 5/8 patients (highly expanded clonotypes with ≥5 isolates seen in three cases: P7, P9, and P12) vs. 2/8 controls (no highly expanded clonotypes) (Table 1). Notably, DQ6-HCRT$_{87-100}$ tetramer

detected significantly more expanded cells in patients (3.51%) than in controls (0.8%), $P = 0.0003$ in a chi-squared test (Table 2).

Expanded TCRαβ clonotypes, albeit mostly using distinct α/β CDR3 sequences, were detected in DQ6$^+$ controls who showed no T1N symptoms. This could reflect T1N-irrelevant or T1N-protective expansion or expansion triggered by non-HCRT epitopes detectable by DQ6-HCRT$_{peptide}$ tetramer due to TCR cross-binding capacity. However, binding to DQ6-HCRT$_{peptide}$ tetramers may not reflect functional cross-reactivity given the discordance between TCR signaling and ligand interaction[22]. In line with the cross-binding hypothesis, a high percentage of expanded clonotypes (34/52, 65.4%) in our dataset were identified by more than one DQ6-HCRT$_{peptide}$ tetramer (Table 1 and Supplementary Data 3). The observed cross-binding capacity was also consistent with the structural similarities in these DQ6/peptide complexes (Fig. 2e). Interestingly, the frequency of expanded DQ6-HCRT$_{HCRT87-100}$ tetramer$^+$ cells in subjects who had received a recent TIV vaccination (C5, C6, C11, P5) was significantly higher than in controls who had received no influenza vaccination (C9, C10) or received H1N1 vaccination ~5 years prior to blood draw during the 2009 flu pandemic (C7, C8, C12), $P = 0.0476$ in the Mann–Whitney $U$-test or $P < 0.0001$ in a chi-squared test. This indicates a high likelihood of detecting cross-binding expanded clonotypes by the DQ6-HCRT$_{HCRT87-100}$ tetramer in the circulating T cell repertoire after recent stimulation. It could be more informative for a T1N analysis to exclude TIV-vaccinated subjects when assessing case/control differences. Indeed, the DQ6-HCRT$_{87-100}$ tetramer detected 10 expanded clonotypes (24 isolates) in 4/7 cases (excluding TIV-vaccinated P5), but none in 5/5 controls (excluding TIV-vaccinated C5, C6, C11), $P < 0.0001$ in a chi-squared test assessing the frequency of expanded cells between patients and controls (Tables 1 and 2). The fact that none of the four patient subjects was previously influenza-vaccinated suggests the epitope in HCRT$_{87-100}$ is relevant to the in vivo expansion of identified clones.

**TRAJ24$^+$ clonotypes isolated by DQ6-HCRT tetramers**. Most clonotypes in our dataset (49/52 expanded or 2460/2465 unexpanded) use unique α/β CDR3s (Supplementary Data 3). Therefore, to investigate public features of expanded clonotypes, we first grouped TCRαβ clonotypes based on their sharing of identical Jα/Jβ genes. Based on the grouping, we focused on those that expressed TRAJ24, the GWAS-identified risk gene[5,6], and that were isolated by the DQ6-HCRT$_{87-100}$ tetramer. We found that, among 57 TCR clonotypes observed more than once in our dataset (containing 302 cells that are either expanded or public, clone ID 1–57 in Supplementary Data 3), there were 9 (I−IX) groups of cells that express identical TRBJ/TRAJ genes although isolated from different subjects (Fig. 4a). These 9 groups covered 3 major types of TCRs (79 cells): Type A (15 cells) contained non-public CDR3s using varied V genes, although J genes are the

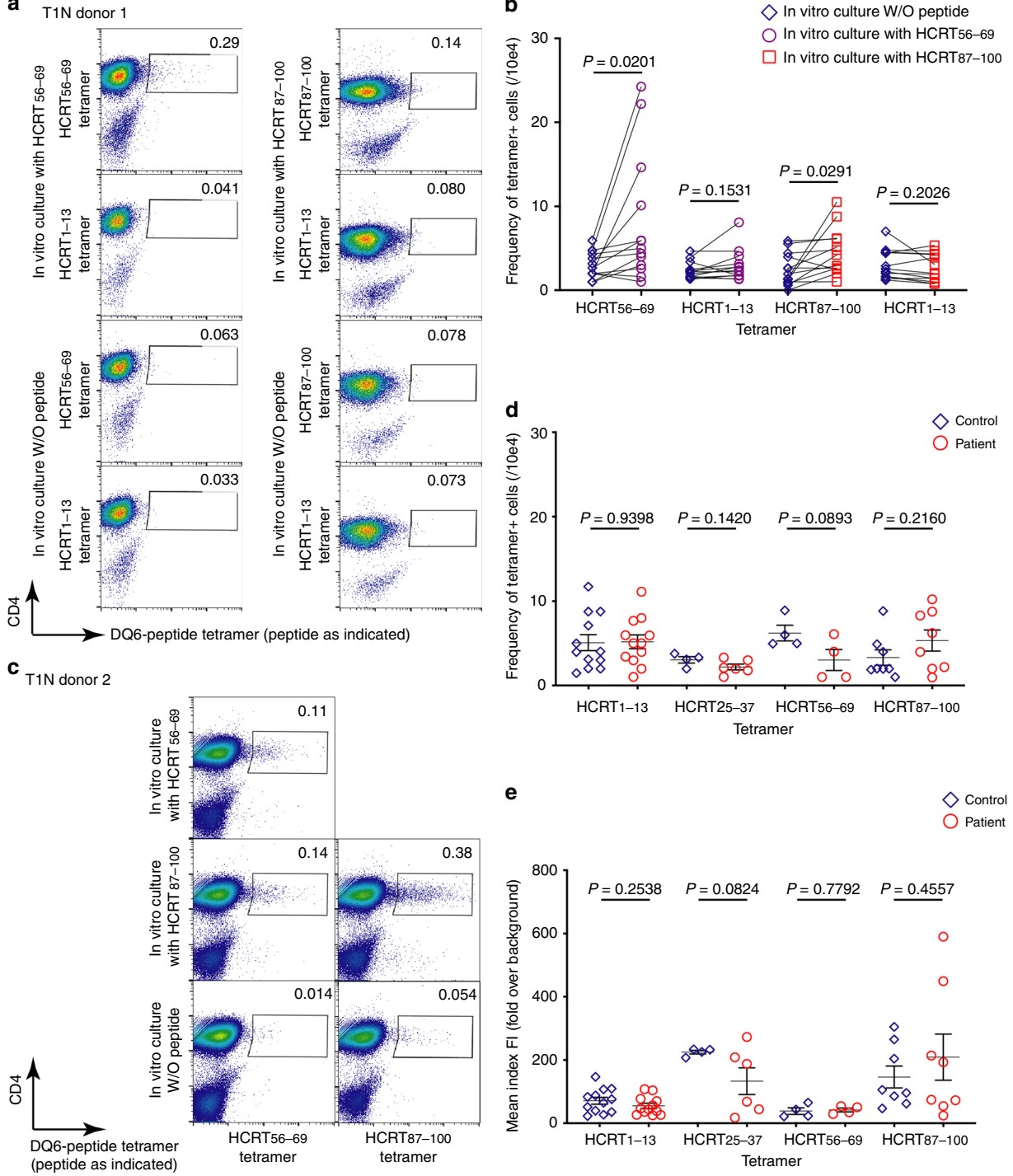

**Fig. 3** Tetramer+/CD4+ T cells in DQ6+ donors. **a** Dot-plots of CD4+ T cells stimulated in vitro with the indicated peptides followed by staining with the indicated tetramers. Frequencies (%) of tetramer+/CD4+ T cells are indicated. **b** Comparison between frequencies of tetramer+ T cells from the same T1N donor (paired as indicated) with and without peptide stimulation. Significance (P < 0.05, bolded) is determined using the paired samples t-test (n = 13 donors). **c** Enrichment (with peptide stimulation vs. without) of DQ6-HCRT56–69 tetramer+/CD4+ T cells in HCRT87–100 peptide stimulated culture. **d** Frequencies or **e** tetramer-binding ranks (mean index fluorescence intensity) of tetramer+ T cells from patients and controls (sample information summarized in Table 1) stained with the indicated tetramers. Statistical analysis compares control and patient samples within each tetramer specificity. Significance (P < 0.05, bolded) is determined using the unequal variance t-test. The error bar represents mean ± SEM

same; Type B (12 cells) used public CDR3α encoded by TRAV10_J18 but different CDR3β encoded by TRBV25–1 in combination with varied Jβ genes, a sequence signature reminiscent of semi-invariant natural killer T (iNKT) cells[41]; Type C (52 cells) used conserved/semi-public α/β CDR3s encoded by

identical V-J genes. 43 type C cells expressed highly conserved clonotypes using TRBV29–1_TRBJ2-5/TRAV6_TRAJ24 genes (42 expanded from C3, P3, and P9; 1 unexpanded public from C12, all in group VIII; Supplementary Data 3). The frequencies of these four genes used in 302 cells were significantly higher from

**Table 1 Summary of single-cell TCR sequencing**

| Subject | Vaccination | Interval vaccination-BD | Tetramer | # Wells | # Cells | # Clonotypes with >1 isolate | # Isolates of each expanded clonotype | Subject | Interval onset-BD | Vaccination | Interval vaccination-BD | Tetramer | # Wells | # Cells | # Clonotypes with >1 isolate | # Isolates of each expanded clonotype |
|---|---|---|---|---|---|---|---|---|---|---|---|---|---|---|---|---|
| **Set A** | | | | | | | | | | | | | | | | |
| C1 | | | L | 96 | 61 | 2 | 4  2 | P1 | 5.7 y | | | L | 96 | 52 | 0 | |
| | | | H1 | 96 | 38 | 1 | 2 | | | | | H1 | 96 | 38 | 0 | |
| C2 | x | 5–6 y | L | 96 | 17 | 0 | | P2 | 4.2 y | x | 5 y | L | 96 | 17 | 0 | |
| | | | H1 | 96 | 27 | 4 | 2  1 | | | | | H1 | 96 | 50 | 0 | |
| C3 | x | 5–6 y | L | 96 | 69 | 4 | **17 11** 7 2 | P3 | 1.6 y | x | | L | 96 | 63 | 4 | **16** 2 1 1 |
| | | | H1 | 96 | 69 | 4 | **10 11** 3 1 | | | | | H1 | 96 | 60 | 3 | **8** 4 2 |
| C4 | x | 5–6 y | L | 96 | 52 | 3 | **10 6** 1 2 | P4 | 4.4 y | x | 5–6 y | L | 96 | 38 | 2 | **2** 2 |
| | | | H1 | 96 | 54 | 3 | **16 5** 3 1 | | | | | H1 | 96 | 55 | 1 | **6** |
| **Set B** | | | | | | | | | | | | | | | | |
| C5 | * | 56 d | L | 96 | 49 | 1 | 3 | P5 | 17.6 y | * | 8 d | L | 96 | 50 | 0 | |
| | | | H2 | 96 | 32 | 1 | 2  2 | | | | | H2 | 96 | 53 | 1 | 2 |
| C6 | * | 7 d | L | 96 | 26 | 2 | 2  2 | P6 | 0.3 y | | | L | 96 | 29 | 0 | |
| | | | H2 | 86 | 51 | 0 | | | | | | H2 | 83 | 58 | 2 | **3** 1 |
| C7 | x | 5 y | L | 95 | 59 | 0 | | P7 | 29.8 y | | | L | 96 | 37 | 2 | **3** 1 1 |
| | | | H2 | 96 | 32 | 1 | 2 | | | | | H2 | 81 | 5 | | 2 1 1 2 |
| | | | | | | | | | | | | H25 | 96 | 43 | 0 | **1** 1 2 |
| C8 | x | 5 y | L | 96 | 50 | 1 | 2 | P8 | 9.7 y | | | L | 96 | 75 | 0 | **1** 1 |
| | | | H2 | 96 | 54 | 0 | | | | | | H2 | 96 | 68 | 1 | 1 2 |
| | | | H25 | 96 | 48 | 0 | | | | | | H25 | 96 | 42 | 0 | |
| C9 | | | L | 96 | 46 | 0 | | P9 | 8.5 y | | | L | 96 | 68 | 4 | **13** 2 2 1 |
| | | | H2 | 96 | 43 | 0 | | | | | | H2 | 96 | 47 | 2 | **3** 2 1 |
| C10 | | | L | 88 | 28 | 0 | | P10 | 11.7 y | | | L | 96 | 48 | 0 | |
| | | | | | | | | | | | | H25 | 96 | 60 | 0 | |
| C11 | * | 141 d | L | 96 | 61 | 2 | 2  1 | P11 | 5.6 y | x | 5.7 y | L | 96 | 46 | 0 | |
| | | | H2 | 87 | 34 | 1 | 1 | | | | | H2 | 96 | 51 | 0 | |
| | | | | | | | | | | | | H25 | 96 | 65 | 0 | |
| C12 | x | 4.75 y | L | 96 | 66 | 2 | 1  2  4 | P12 | 3.3 y | x | | L | 96 | 54 | 0 | |
| | | | H2 | 96 | 43 | 1 | 1 | | | | | H2 | 87 | 48 | 1 | **11** 1 1 |
| | | | H25 | 96 | 47 | 0 | | | | | | H25 | 96 | 51 | 2 | **4** 2 1 |
| | | | | 96 | 43 | 0 | | | | | | H25 | 96 | 39 | 0 | **1** |
| | | | | 96 | 28 | 0 | | | | | | | 96 | 45 | 0 | |
| Total | | | | | | | 146 | | | | | | | | | 143 |

Subject numbers indicate the order of pairs (C: control; P: patient) used in blinded experiments. "x" indicates vaccination. "*" indicates vaccination using Pandemrix during the 2009 flu pandemic. d: day. y: year. """ indicates seasonal trivalent inactivated influenza vaccination (TIV) prior to blood draw (BD). "# Wells" counts single CD4+ T cells sorted by the indicated tetramer for sequencing analysis. "# Cells" counts sequenced cells with fully identified TCRαβ gene information and productive CDR3 sequences. "# Clonotypes with isolates >1" counts expanded TCRαβ clonotypes. Isolates of an expanded clonotype counts cells with identical α/β CDR3 sequences. Identical clonotypes identified by different tetramers (expanded cross-binding clonotypes) are aligned, but the alignment across subjects does not reflect related TCRs. Highly expanded clonotypes (≥5 total isolates per subject) are indicated in bold. Tetramer specificities are as follow, L: HCRT$_{1-13}$; H1: HCRT$_{56-69}$; H2: HCRT$_{87-100}$; H25: HCRT$_{25-37}$. Elucidation of functionality and the triggers for highly expanded cross-binding clonotypes found in controls (e.g., C3 and C4) are beyond the scope of this study

**Table 2 Comparison of expanded clones from controls and patients analyzed using the chi-squared test**

| DQ6 tetramer | Controls expanded/total cells (% cells expanded) | Patients expanded/total cells (% cells expanded) | Chi-squared test (P-value) | FDR correction | |
|---|---|---|---|---|---|
| | | | | q-value | Discovery? (P < 0.047619) |
| All | 134/2660 (5.04) | 117/2843 (4.12) | 0.1013 | 0.063819 | No |
| HCRT1–13 | 74/1141 (6.49) | 71/1143 (6.21) | 0.7884 | 0.41391 | No |
| HCRT25–37 | 2/384 (0.52) | 0/576 (0) | 0.0829 | 0.063819 | No |
| HCRT56–69 | 52/384 (13.54) | 20/384 (5.21) | <0.0001 | 0.000156 | Yes |
| HCRT87–100 | 6/751 (0.8) | 26/740 (3.51) | 0.0003 | 0.000315 | Yes |
| *Excluding TIV-vaccinated subjects* | | | | | |
| HCRT87–100 | 0/463 (0) | 24/644 (3.73) | <0.0001 | 0.000156 | Yes |

Statistics was corrected with the Benjamini, Krieger and Yekutieli false discovery rate (FDR)-controlling procedure, desired FDR (Q): 5%. q-value: adjusted P-value. Discovery indicates significant skewing of expanded clonotype detection in one group versus another. Analysis excluding TIV-vaccinated subject is performed on DQ6-HCRT$_{87-100}$ tetramer-sorted cells

those in 2762 cells with determined α/β CDR3s or in 4605 wells that returned TCR sequences (Fig. 4b). Therefore, the skewing of their usage in these 302 cells, for example, a 5.65-fold increase of TRAJ24, was unrelated to sequencing bias, but rather a reflection of biological relevance.

We then used the GLIPH algorithm[24] to group α/β CDR3s based on their sharing of common sequence motifs (global, local, or single as previously defined[24]). These motifs (Supplementary Data 4) likely determine the antigen binding specificity of TCRs in the corresponding groups. We found 25/204 CDR3β motifs and 16/941 CDR3α motifs containing significantly enriched common V genes and expanded clones (Fig. 5a). Although all cells were sorted by tetramer specificity, it was uncommon for expanded clonotypes from multiple subjects to use public β and α CDR3 motifs (from the 25 β and the16 α motif groups, Fig. 5b). Our analysis showed that, only once did clonotypes using β motif (global-E%DRGRSET, % = varied residues) and α motif (global-%TDSWGK) occur in three subjects (P3, P9, C12) and three clonotypes were found to be in vivo expanded in the two patients (Fig. 5c). In addition, all of these clones expressed TRBV29-1_TRBJ2-5/TRAV6_TRAJ24 genes. Collectively, both analyses on the sharing of Jα/Jβ genes and β/α CDR3 motifs suggest unique public features of TRBV29-1_TRBJ2-5/TRAV6_TRAJ24 expressing clones.

Considering the T1N genetic risk conferred by the TRAJ24 gene[5,6], we next focused on all TRBV29-1_TRBJ2-5/TRAV6_-TRAJ24 expressing clones. We found 44 such clones out of 74 TRAJ24$^+$ cells in our dataset (Fig. 5c). All α chains used conserved CDR3 sequences, CALxxDSWGKF(L)QF. For all TRAJ24-bearing CDR3α sequences from our dataset, the G/C SNP alleles were always in frame with two thymidines in the codon TT<u>G</u> or TT<u>C</u>, encoding L/F (Supplementary Fig. 6a). However, in vivo expansion of TCRαβ clonotypes carrying the L variant was only observed in the two cases (P3 and P9) out of the 24 individuals tested in this study. These expanded clonotypes (eTRAJ24L hereafter) shared similar binding features, as they were isolated by DQ6-HCRT$_{1-13}$ and DQ6-HCRT$_{56-69}$/HCRT$_{87-100}$ tetramers (epitopes in HCRT$_{56-69}$ and HCRT$_{87-100}$ share high homology). Notably, although expanded, the clonotype using TRBV29-1_TRBJ2-5/TRAV6_TRAJ24 genes in C3 used a less conserved CDR3β (without E%DRGRSET motif) paired with the TRAJ24F-bearing CDR3α. In addition, the highly expanded eTRAJ24L clonotype from patient 9 (TCR27 hereafter) was composed of a public CDR3α observed in P7, P8, P9, P12, and C12; and a public CDR3β observed in P8, P9, P10, C11, and C12 (Supplementary Fig. 6). The only other isolation of this public TCRαβ clonotype was one unexpanded cell from C12 isolated by the DQ6-HCRT$_{25-37}$ tetramer (Fig. 5c). Notably, index

FI analysis showed an intermediate tetramer-binding rank for cells expressing eTRAJ24L clonotypes (Fig. 5d), as might be expected for a self-reactive T cell, which survived thymic selection through low to moderate TCR affinity[42] and then expanded in vivo.

**A TRAJ24-G allele$^+$ TCR signals after binding to HCRT$_{87-100}$.** To further examine whether binding to HCRT$_{87-100}$ triggers the signaling of eTRAJ24L clonotypes, we expressed TCR27 in a TCRαβ-deficient Jurkat cell line. We also generated control transfectants for comparison: one with TRAJ24$^{neg}$ TCR26 from patient P12, which is a highly expanded and cross-binding clonotype that lacks public features; one with a T1N-irrelevant TCR isolated from a CD8$^+$ T cell; others with tetramer-identified TRAJ24$^{neg}$ TCRs bearing iNKT-like signatures (Fig. 6a and Supplementary Fig. 7a). We reconstructed TCRs using sequence-determined α/β CDR3 nucleotides in frame with the germline sequences of the identified gene segments (IMGT/V-QUEST[43]). Flow cytometric analysis confirmed the surface expression of these TCRs in the Jurkat transfectants (Fig. 6b and Supplementary Fig. 7d). Compared to the irrelevant control, transfectants with TCR27, TCR26, and iNKT-like TCRs showed more TCR$^+$/ tetramer$^+$ cells and relatively higher staining signals of HCRT$_{1-13}$ or HCRT$_{87-100}$ tetramers (Fig. 6c, d and Supplementary Fig. 7c, e, f), consistent with the initial tetramer isolation of single cells expressing these TCRs. The moderate difference from control (~1.5–2.5 fold in MFI) was consistent with the intermediate rank of these TCRs, as determined by iFACS (Fig. 5d and Supplementary Fig. 3).

We next tested activation of these TCR transfectants by DQ6-restricted presentation of a HCRT epitope. TCR transfectants were incubated with control stimuli or HCRT peptides presented by an antigen-presenting cell (APC) line that expresses DQ6 as the only HLA class II molecule. All transfected TCRs transmitted a CD3/CD28-mediated signal, indicating cell surface, functional CD3 co-expression with TCR. Specifically, DQ6-restricted presentation of HCRT$_{87-100}$ peptide to TRAJ24$^+$ TCR27 triggered moderate TCR-mediated signaling, out of all HCRT peptide/TCR pairs tested using the validated Jurkat-reporter system[24] (Supplementary Fig. 7g, h). As a physiological amidation process converts the C-terminal Gly98 of HCRT2 neurotransmitter to a C-terminal amide (-NH2)[35], the likelihood of HCRT$_{87-97}$-NH2 being presented by DQ6 in vivo may be higher than HCRT$_{87-100}$. Consistent with this hypothesis, we found that HCRT$_{87-97}$-NH2 triggered stronger signaling in TCR27 transfectants than did HCRT$_{87-100}$ (Fig. 6e and Supplementary Fig. 7i). This result indicates that the epitope in HCRT$_{87-97}$-NH2 is capable of triggering the in vivo clonal expansion of TCR27$^+$ cells.

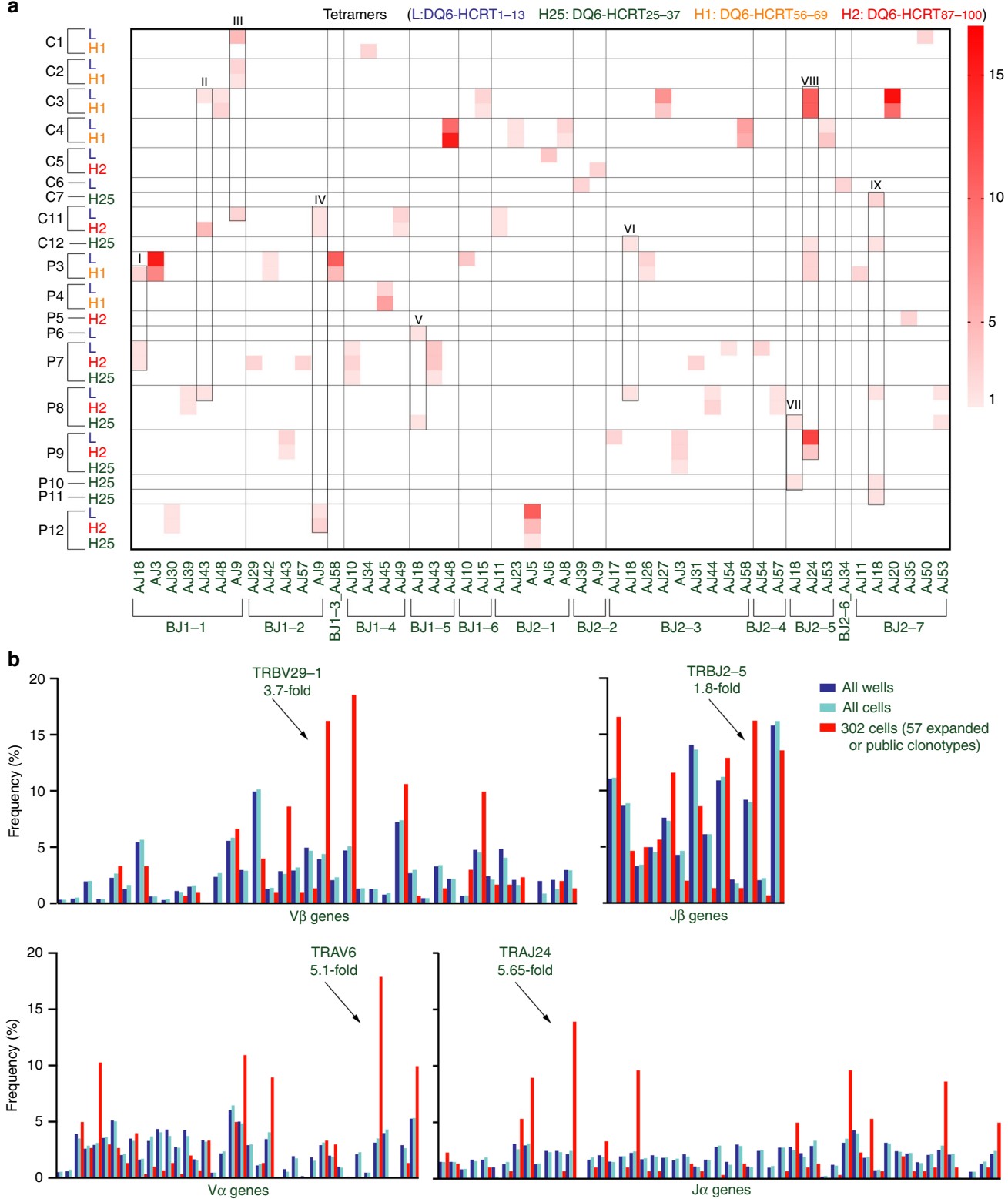

**Fig. 4** Public features of clonotypes with more than one isolate from our dataset. **a** A heat map illustrating 9 groups (I–IX) of cells from multiple subjects sharing identical paired TRBJ/TRAJ genes. Number of isolates of the same clonotype is proportional to the color intensity. Tetramer specificity is differentiated by colors. Tetramer categories without any hit on the heat map are not shown. **b** Frequencies of V or J genes used in 57 TCRα/β clonotypes with multiple isolates compared with "All Wells" that return sequences (see Supplementary Data 2) or "All Cells" with productive α/β CDR3 sequences (see Supplementary Data 3). Arrows indicate genes used in TRBV29–1_TRBJ2-5/TRAV6_TRAJ24-expressing cells, with fold increase compared to "All Cells"

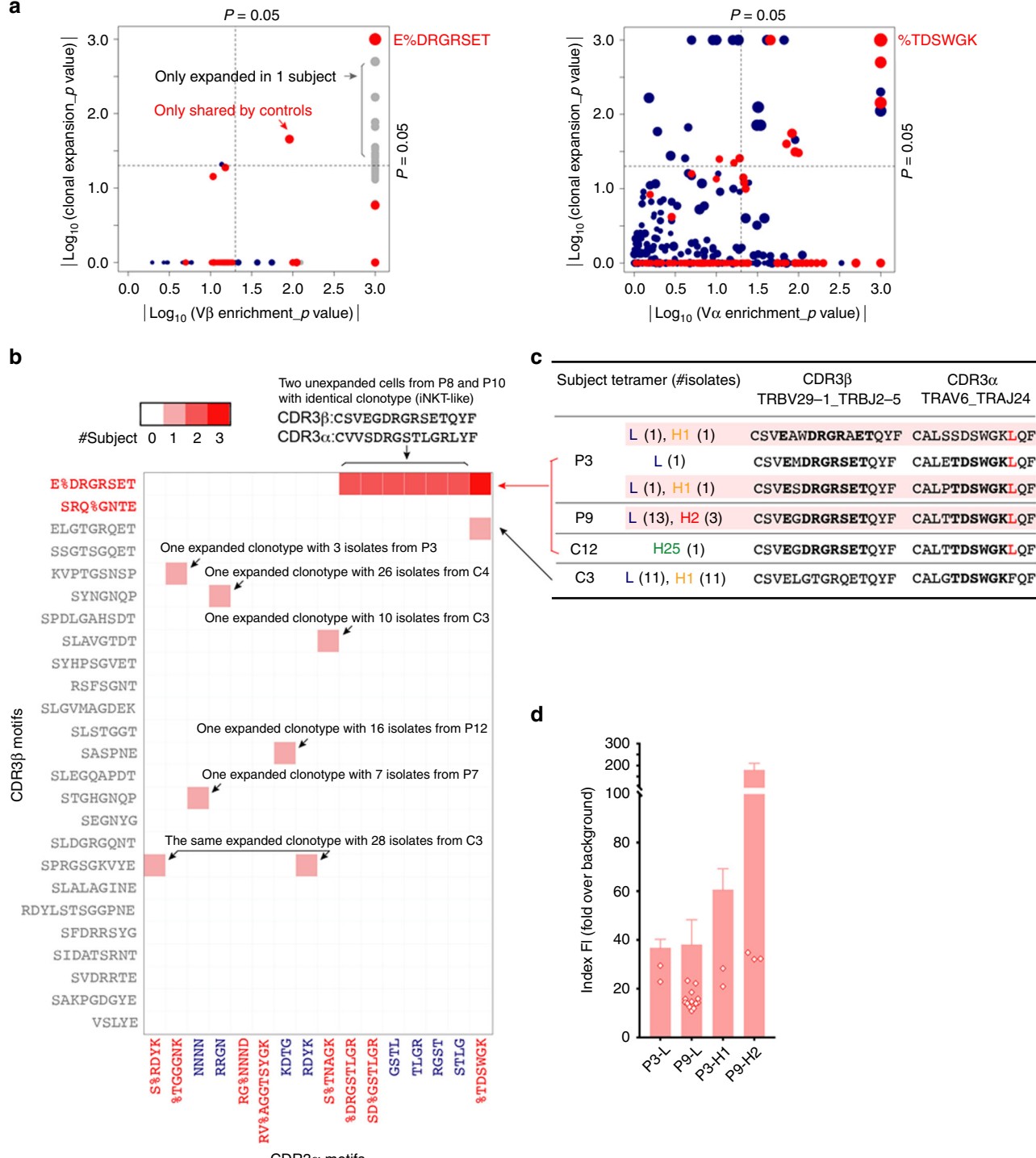

**Fig. 5** Public features of the eTRAJ24 clonotypes. **a** Analysis of CDR3 motifs shared by groups of TCRβ (left) or TCRα (right) by GLIPH algorithm[24]. Each dot represents one motif: global (red), local (blue), and single (gray) (see Supplementary Data 4 for definitions). 25 CDR3β motifs and 16 CDR3α motifs reach significance ($P < 0.05$, the upper right quadrant) when analyzed for enrichment of V genes and expanded clones compared to a reference dataset as described[24]. The α/β motifs used in eTRAJ24 clonotypes (shown in red to the right of each plot) have the highest overall significance. The radius of dots reflects the final score (see details in Supplementary Data 4). **b** Number of subjects with clonotypes using paired α/β motifs from the 25 β and 16 α motifs. Motifs are color-coded as in **a**. #clonotype with #isolates from the indicated donor(s) is also shown on the heatmap or summarized in **c**. **c** Summary of tetramer-identified clones expressing TRBV29-1_TRBJ2-5/TRAV6_TRAJ24 genes. The L residue of the G (SNP) allele is indicated in red. The three eTRAJ24L clonotypes are shaded. The CDR3 sequences using E%DRGRSET and/or %TDSWGK motifs are bolded. **d** Tetramer binding rank of individual eTRAJ24L+ cells. Bar graphs represent the mean ± SEM FI of all sorted CD4+/tetramer+ single cells from the indicated subject-tetramer category. Open symbols are for only eTRAJ24L+ cells within the same subject-tetramer category, as in the corresponding bar graph. All values are normalized with background MFI of the entire CD4+ population in the tetramer detection channel

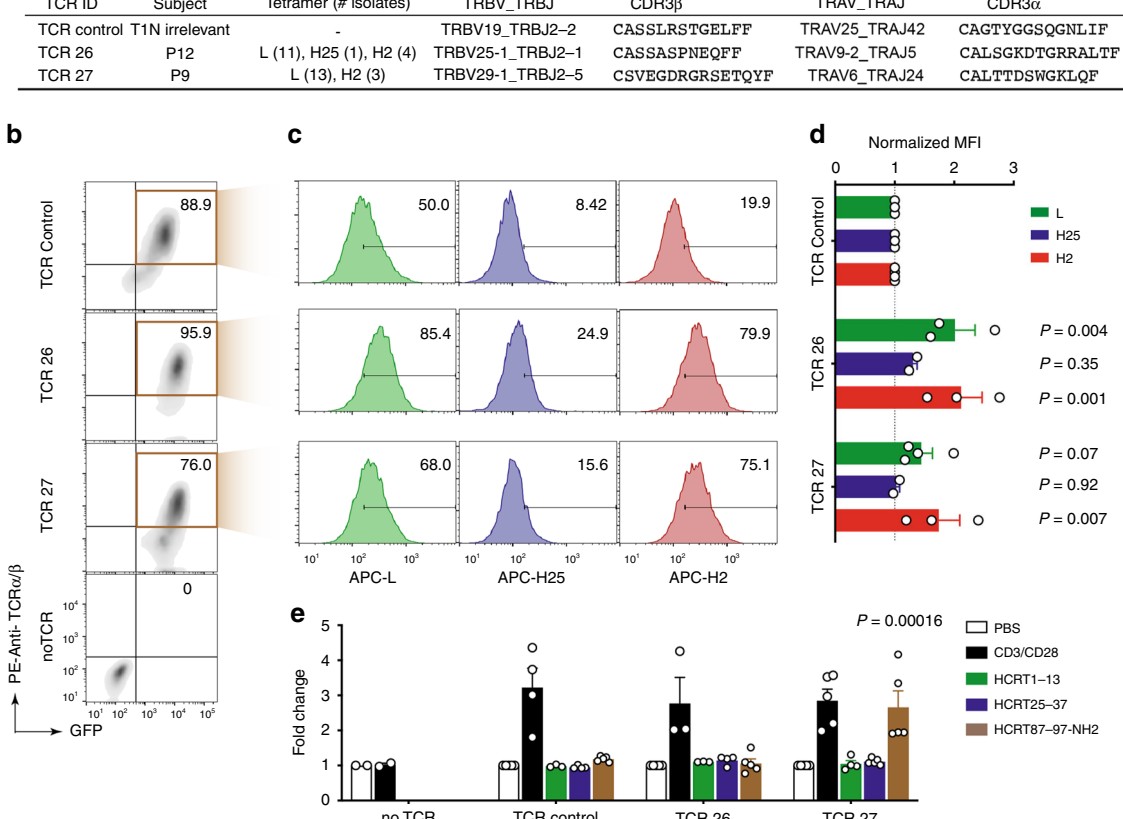

**Fig. 6** Validation of tetramer binding and TCR signaling. **a** TCRs tested by transfection into Jurkat-luciferase reporter system[24]. **b** Dot-plots of Jurkat transfectants expressing TCRs as shown in **a**. Expression of GFP indicates a successful transfection of plasmids directing the expression of TCRs. **c** Tetramer staining of TCR+ Jurkat cells with percentages of tetramer+ cells indicated. **d** MFIs of the tetramer-associated signal on the surface of TCR+ cells are compared. **e** Jurkat cells expressing the indicated TCR were incubated with DQ6-expressing APCs (HLA class II- K562 cells transfected with DQ6) in the presence of HCRT peptides or control stimuli: PBS (negative) or anti-CD3/anti-CD28 Abs (positive). Luciferase activity reflecting TCR-mediated signaling was measured and analyzed as previously described[24]. The bar chart shows mean ± SEM, with the corresponding data points overlaid. Significance ($P < 0.05$, bolded) is determined using $t$-test comparing the indicated TCR and the TCR control

**Phenotypic features of expanded TRAJ24-G allele+ cells.** The single-cell sequencing pipeline is particularly useful for determination of phenotypic features of expanded clones by TCR clustering[23]. Before determination of phenotypes of eTRAJ24L+ cells, we first analyzed the skewing of transcripts for 25 tested transcriptional factors and cytokines within expanded versus unexpanded cells. The frequencies of expanded clones positive for TBX21 (252/283, 89% vs. 419/2295, 18.3%; by 4.8 fold), IFNγ (107/293, 37.8% vs. 54/2295, 2.4%, by 16 fold), or PRF1 (146/283, 51.6% vs. 108/2295, 4.7%; by 11 fold) were significantly ($P < 0.0001$, chi-squared test) higher than that of the corresponding transcript+ unexpanded clones (Fig. 7a, b). Notably, the in vivo transcriptional features of expanded clones, which are associated with ongoing or prior immune responses, did not distinguish all tetramer+/CD4+ T cells in patients compared to controls, as this comparison was heavily influenced by the large portion (2295 unexpanded vs. 283 expanded; 8 fold) of unexpanded cells (Supplementary Fig. 8a and Supplementary Data 5). Consistent with the significantly higher numbers and/or more isolates of expanded clones from six patients (P3, P4, P7–9, and P12) and three vaccinated controls (C3, C4, and C11) (Table 1), TBX21 (encoding T-bet) and PRF1 (encoding perforin) were found more frequently in these nine donors (Supplementary Fig. 8b and Supplementary Data 5). This expression pattern supports a previously proposed immune mechanism in which the in vivo

expansion of perforin-expressing CD4+ T clones with cytotoxic potential was dependent on T-bet[44].

Because the above phenotypic features are unique to expanded clonotypes, a very similar difference was observed when the 21 eTRAJ24L+ cells found in patients were compared to 16 unexpanded TRAJ24+ cells found in controls or to 21 unexpanded TRAJ24L+ cells in all DQ6+ individuals (Fig. 7c, d). In line with the T-bet+ effector phenotype (95.2% TBX21+) for the 21 eTRAJ24L+ cells is the extremely low-frequency detection of RORC (0), FOXP3 (0), and BCL6 (1/21) mRNAs. Consistent with lack of FOXP3, expression levels of other regulatory T (Treg) cell surface markers, CD25 and CD127, on expanded clones, including the 21 eTRAJ24L+ cells, showed no difference from the levels on the tetramer-neg/CD4+ population (Supplementary Fig. 9). Further, the solo TCR27 clonotype isolated from C12 (Fig. 5c) had 0 phenotypic transcripts (Fig. 7c). These phenotypic findings are consistent with the sequencing results indicating in vivo expansion of eTRAJ24L+ cells.

A TRAJ24F+ clone from C3 also expanded in vivo into effectors (100% TBX21+ vs. 12.5% in the eight unexpanded TRAJ24F+ cells). However, these 22 cells less frequently expressed PRF1 (40.9% vs. 81.0%, $P = 0.0073$, chi-squared test) and TGF-β (18.2% vs. 71.4%, $P = 0.0004$), but more frequently expressed IFNγ (77.3% vs. 9.5%, $P < 0.0001$) than the 21 eTRAJ24L+ cells from patients (Fig. 7c, d and Supplementary

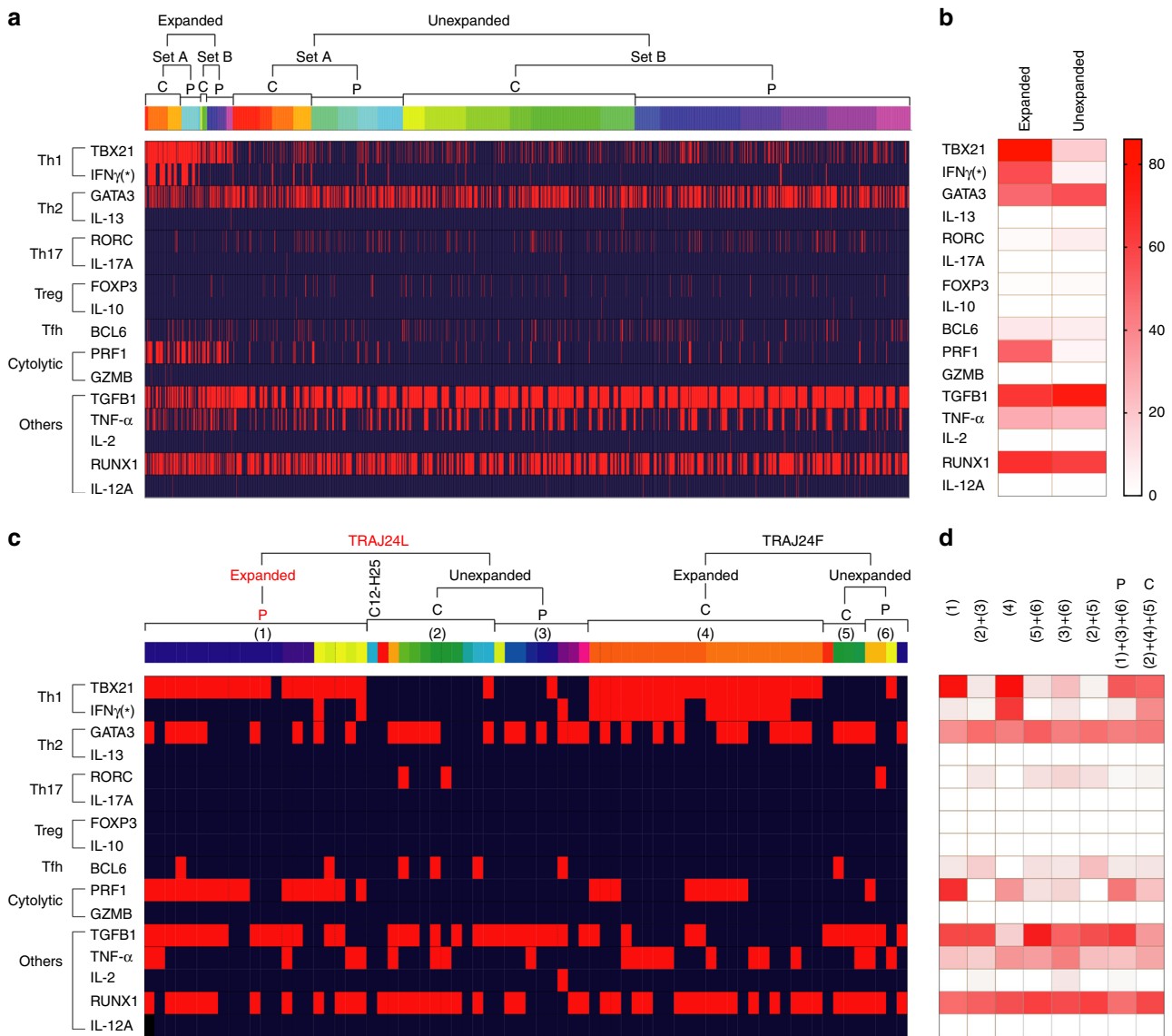

**Fig. 7** Phenotypic features of tetramer+/CD4+ cells expressing the TRAJ24 gene. **a** Sequenced cells from controls (C) or patients (P) in each set (Table 1) are clustered as indicated. Each color in the horizontal bar above the heat map represents one donor. Red indicates the presence of mRNA in the cell. Genes to the left of the heat map are grouped according to their usage in different conventional T cell subsets[23]. *Different sets of primers were used to amplify IFN-γ mRNAs in cells from Set A and Set B donors. **b** Frequencies of expanded versus unexpanded cells expressing the indicated transcript. **c** Phenotypes of expanded or unexpanded TRAJ24L+ clones versus TRAJ24F+ clones. Each color in the horizontal bar represents one subject-tetramer category of single cells. Patients (P) or controls (C) are clustered as indicated. The expanded TRAJ24L (red font) cluster includes the unexpanded clonotype from P3, as it only differs from other eTRAJ24L clonotypes from the same patient by 1 residue in both α/β CDR3 sequences. **d** Frequencies of cells from the indicated cluster expressing the indicated transcript in **c**. The scale of color intensity is the same as in **b**

Data 6). This indicates that eTRAJ24L+ cells acquired a phenotype that differs from conventional $T_{h1}$ cells. The phenotypic difference between TRAJ24+ cells in patients and those in controls (more PRF1, $P = 0.0105$, and TGF-β, $P = 0.0016$, but less IFNγ, $P = 0.0007$ in patient cells) was largely attributed to differences observed in expanded TRAJ24+ clones (Fig. 7d and Supplementary Data 6). Collectively, these data suggest that DQ6-restricted eTRAJ24L+ cells in T1N patients have undergone in vivo expansion and acquired an unconventional effector phenotype with cytotoxic potential, likely after recognition of epitopes in $HCRT_{56-69}$ or $HCRT_{87-100}$.

## Discussion

In this study, we focus on using in vivo clonal expansion together with key genetic factors and molecular signatures to demonstrate

the existence of self-reactive CD4+ T cells and link their expansion to a likely autoantigen. Tetramer staining offers an unbiased antigen-specific approach to isolate cells of interest, which must then be further assessed to pinpoint truly autoreactive T cells among tetramer+ candidates. In particular, the discordance between tetramer binding and TCR signaling, proved here in self-antigen-binding T cells and previously in foreign-antigen-binding T cells[22,24,28–30], indicates that direct measurement of the signaling capacity is critical in determination of function.

The ex vivo single-cell analysis uniquely allows determination of in vivo clone sizes by pairing of TCRαβ and also allows linkage of clonotype with informative phenotypic characteristics. Given the substantial diversity in TCRαβ clonotypes[24], the low frequency of tetramer+ cells, the likely involvement of polyclonal T cells in T1N, and the potential heterogeneous pathways that lead

to autoimmunity among different individuals, it is challenging to directly isolate public TCRαβ clonotytpes shared by multiple individuals. However, we identified a family of TCRαβ clonotypes expressing TRBV29-1_TRBJ2-5/TRAV6_TRAJ24 genes from among thousands of ex vivo accessible DQ6-HCRT$_{peptide}$ tetramer$^{+}$/CD4$^{+}$ cells from DQ6$^{+}$ individuals with/without T1N. Notably, this family of clonotypes observed in 4/24 donors from our study is absent from previous in vitro studies[7,21]. Within our dataset, in vivo expansions (determined by sequence and phenotype) of G-allele$^{+}$ eTRAJ24L clonotypes are only observed in T1N patients (2/12) from the 24 donor blood samples. Related clonotypes are absent in the other patient donors, likely due to both heterogeneous biology facts and technical limitations, such as the low number of cells sequenced/donor. Nonetheless, we found clear differences between eTRAJ24L$^{+}$ cells in patients and related cells in controls: (1) the solo isolate sharing identical α/β CDR3s with TCR27 from a DQ6$^{+}$ control has not expanded, and (2) the expanded TRAJ24F$^{+}$/CD4$^{+}$ T cells isolated from another DQ6$^{+}$ control possess different effector phenotype compared to eTRAJ24L$^{+}$ cells. We cannot rule out a potential role of the TRAJ24F$^{+}$ clonotype in T1N, although these cells are from a control. Indeed, the HCRT$_{87–97}$-NH2 (SGNHAAGILTM-NH2) epitope that stimulates TCR 27 signaling is likely physiologically available for DQ6-restricted presentation in all DQ6$^{+}$ individuals. This implies that other factors are necessary to trigger T1N development in DQ6+ individuals who carry the risk TRAJ24 gene.

Although antigen presentation in the central nervous system (CNS) is not yet well-understood, a recent study has suggested that conventional dendritic cells are essential for presenting CNS-derived antigens and licensing T$_{h}$ cells to initiate neuroinflammation[45]. It is possible that the unconventional CD4$^{+}$ T$_{eff}$ cells that we identified receive licensing via a similar MHCII (DQ6) presentation of NHAAGILTL(M) epitopes, and then activate microglia to secrete neurotoxic factors or exert their own cytotoxic potential to destroy microglia. Either process would be detrimental to the associated neurons[46]. In addition, fragments of HCRT proteins that include the epitopes may leave the CNS, and prime T cells outside of the brain, similar to a recent murine model in which insulin peptides released by pancreatic β-cells initiate diabetic T cell responses at distant lymph nodes[47]. Our finding of eTRAJ24L clonotypes in T1N patient PBMCs indicate the presence and persistence of brain tissue-reactive TCR clonotypes in the circulation, reminiscent of recent evidence that gluten-specific TCR clonotypes persist in blood and overlap with clonotypes in gut biopsies from celiac patients[40].

Many expanded clonotypes including eTRAJ24L bind to more than one DQ6-HCRT$_{peptide}$ tetramer. TCRs can recognize class II bound to peptides that share certain homology but differ at some TCR-facing residues[26], likely due to the ability of a TCR to use various modes of ligand interaction[48]. Our observation either reflects a general characteristic of self-reactive T cells or implies a unique promiscuous feature of DQ6-reactive TCRs. The former is consistent with the finding that many memory T cells express cross-reactive TCRs[26]. The latter is supported by the NetMHCIIpan[33] prediction that alanine is preferred at all core residues of DQ6-binding peptides that lack side chains to increase TCR specificity. Indeed, all five regions of prepro-HCRT that generate DQ6-binding peptides contain multiple alanine residues. The cross-reactive potential suggests a possible molecular basis for promotion of T1N by viral proteins during the 2009 flu pandemic[49,50], for example by mimicking HCRT epitopes, as suggested[7], although functional cross-reactivity by TCRs from T1N patients still requires further proof.

The promiscuous feature of DQ6-reactive TCRs has unexpectedly complicated current studies (inducing ours) that are focused on the identification of self-antigen and truly autoreactive T cells. It is currently unclear whether expanded multi-tetramer-binding clonotypes other than eTRAJ24L (including the TRAJ24F$^{+}$ clone found in a control) mediate ongoing immune responses or not, and whether these in vivo expansions are associated with (including regulatory function) or irrelevant to T1N. Because tetramer$^{+}$ cells do not necessarily signal in response to the HLA-peptide ligand[22,24], multi-tetramer-binding also does not ensure signaling to both epitopes, as suggested by the null function of TCR27 transfectants in response to HCRT$_{1–13}$, and the failure of TRAJ24$^{neg}$ TCR clonotypes-mediated signaling in response to all tested HCRT peptides. The physiological significance of thymic selection for many self-binding but non-autoreactive cells may be to maintain self-tolerance, for example by selectively sequestering DQ6-HCRT complexes, which would otherwise trigger signaling-competent T cells for autoimmunity. Also notable is that TRAJ24 gene can be rearranged with various Vα genes and choose from extremely diverse β chain genes for pairing in different T cells from different DQ6$^{+}$ individuals, as observed herein and previously[7,24]. Therefore, polyclonal TCRs using distinct TRAJ24L-bearing CDR3α sequences or factors other than TCRs may be involved in T1N development in different individuals. Nonetheless, our discovery of eTRAJ24L clonotypes offers a candidate TCR for further investigation of autoimmunity in T1N.

The two homologous epitopes at the C-termini of HCRT1 (NHAAGILTL) and HCRT2 (NHAAGILTM) may make different contributions to autoimmunity rather than redundantly boosting the response. Indeed, HCRT1 and HCRT2 proteins differ in stability and binding affinities to HCRTR1 and HCRTR2 receptors[35], with possible consequences for tolerance. Our findings thus raise the next set of mechanistic questions, while uncovering molecular linkage between autoimmune effectors and targets in T1N.

## Methods

**Construction of recombinant HLA–DQ6-HCRT complexes**. A stable Drosophila Schneider 2 (S2) insect cell line secreting soluble DQ6 proteins was previously constructed[31]. In this construct, the class II-associated invariant chain peptide, CLIP$_{87–101}$ (aa: PVSKMRMATPLLMQA), is covalently linked to the β chain of DQ6 α/β heterodimers. The DQ6 construct includes the extracellular portion of HLA-DQA1*01:02 followed by a 3C protease cleavage site and leucine zipper-Fos sequence and the extracellular portion of HLA-DQAB1*06:02, preceded by the peptide sequence and a thrombin-cleavable linker sequence (GGGGGSLVPRGSGGGG), and followed by a 3C protease cleavage site and leucine zipper-Jun sequence. Similarly, we constructed five S2 cell lines expressing soluble DQ6-HCRT$_{1–13}$ (aa: MNLPSTKVSWAAV), DQ6-HCRT$_{25–37}$ (aa: ALLSS-GAAAQPLP), DQ6-HCRT$_{56–69}$ (aa: AGNHAAGILTLGKR), DQ6-HCRT$_{87–100}$ (aa: SGNHAAGILTMGRR) and DQ6-EBV$_{486–499}$ (aa: RALLARSHVERTTD), respectively. Briefly, two plasmids (encoding α and β chains of DQ6) were used for the expression of each DQ6–peptide complex in S2 cells. The α chain-encoding plasmid is shared by all constructs, and the β chain-encoding plasmids were modified via polymerase chain reactions (PCR) and sub-cloning to swap the nucleotide sequence encoding corresponding peptides that were covalently tethered to the N-terminus of DQ6β. S2 cells were co-transfected following the user guide for Drosophila S2 cells (Invitrogen, Thermo Fisher Scientific) with α-encoding and β-encoding plasmids as well as a third plasmid carrying the neomycin (geneticin)-resistance gene, at a ratio of 20:20:1. Geneticin (G418)-resistant S2 transfectants were recovered after 2–3 weeks of culturing in Schneider Drosophila medium with 10% heat-inactivated fetal bovine serum (HI FBS), 2 mM glutamine and 1.5 mg/ml G418 (all from Thermo Fisher Scientific). Stable cell lines were established after another 2–3 weeks of culturing and selection under G418.

**Expression and purification of soluble HLA proteins**. Stable S2 cell lines secreting soluble HLA proteins (e.g., a DQ6-HCRT$_{peptide}$ complex or DM[31]) were initially cultured in the complete Schneider medium and gradually adapted to S2 serum-free medium (Thermo Fisher Scientific) before the induction of protein expression using 1 mM copper sulfate. After 1-week induction, bacteriostatic protease inhibitors, such as 1 mM phenylmethane sulfonyl fluoride (PMSF), 1 mM ethylenediaminetetraacetic acid (EDTA), and 0.02% sodium azide (NaN$_3$), were added to the S2 culture, which was then centrifuged to collect supernatants containing soluble HLA proteins. The 0.22 micron membrane-filtered supernatant was then applied onto a column for the purification of target proteins by affinity

chromatography. A customized anti-DQ column containing SPV-L3 Ab[31] was used to purify DQ6 and a column composed of M2 (anti-FLAG tag) resins (Sigma) was used to purify DM. Affinity-purified proteins were further concentrated and isolated from aggregates or degraded material by size-exclusion chromatography, using either Superdex increase 200 10/300 GL or HiLoad 16/60 Superdex 200 gel filtration columns (GE Healthcare). Fractions were eluted with TBS buffer (e.g., 20 mM Tris–Cl, 150 mM NaCl, pH 7.4) and the ones containing monomeric forms of each protein were pooled. Protein purity was confirmed using Coomassie and western blotting analyses, and protein functionality was validated in the peptide-binding assay, as described below.

**Peptide competition assay.** The ability of a peptide to inhibit the interaction of DQ6 and a reference binding peptide at steady state was used to estimate the relative DQ6-binding capacity of test peptides. We used biotinylated $EBV_{486–500}$ (aa: biotin-GGGRALLARSHVERTTDE, synthesized by Genscript), a DQ6-binding peptide (epitope underlined) derived from Epstein-Barr virus nuclear antigen[32], as our reference peptide. Non-biotinylated test peptides included 30 15-mer overlapping peptides derived from prepo-HCRT (Supplementary Data 1a, by Genscript) and the positive control peptide $EBV_{486–500}$. The DQ6-$CLIP_{87–101}$ construct contains a thrombin cleavage site in between $CLIP_{87–101}$ and DQ6β. To cleave the covalent linker and enable replacement of $CLIP_{87–101}$ by high-affinity DQ6 binders, soluble DQ6-$CLIP_{87–101}$ at a concentration of 3 μM was incubated with 0.002 U/μl thrombin enzyme (Novagen, EMD Millipore) for 2 h at room temperature (RT) prior to peptide loading experiments. To test DQ6-binding capacity, a non-biotinylated peptide at 40 μM was mixed with 1 μM biotinylated $EBV_{486–500}$ and incubated with 25 nM thrombin-cleaved DQ6-$CLIP_{87–101}$. 100 nM soluble DM was added as a catalyst to increase the peptide exchange efficiency[31]. The reaction was carried out under acidic conditions in 100 mM acetate buffer (acetic acid and sodium acetate, pH 4.6), 150 mM NaCl, 1% (w/v) BSA, 0.5% (v/v) IGEPAL CA-630 (Sigma), 0.1% (w/v) NaN₃, at 37 °C for 20 h. After incubation, the peptide exchange reaction was stopped by the addition of two volumes of the neutralization buffer [100 mM Tris–Cl (pH 8.3), 150 mM NaCl, 1% (w/v) BSA, 0.5% (v/v) IGEPAL CA-630, 0.1% (w/v) NaN₃], and the mixture was transferred to an SPV-L3-coated 96-well plate and incubated at RT for 1 h. Time-resolved fluorescence representing DQ6-associated biotinylated $EBV_{486–500}$ captured by SPV-L3 in each well was then quantified using the DELFIA Eu-N1 Streptavidin System (PerkinElmer).

**DQ6-HCRT crystallization and structure determination.** DQ6-$HCRT_{56–69}$ proteins purified from S2 culture were incubated with recombinant HRV 3C protease (3C^pro, Novagen, EMD Millipore) at 4 °C overnight to remove the leucine zipper at the C-termini of DQ6α/β heterodimers. 3C^pro-cleaved DQ6-$HCRT_{56–69}$ was further purified by anion exchange chromatography using HiTrap Q HP and finally by gel filtration using HiLoad 16/60 Superdex 200 (GE Healthcare). For crystallization, DQ6-$HCRT_{56–69}$ was concentrated to 10 mg/ml in 20 mM Tris–Cl pH 7.5, 20 mM NaCl, 0.01% NaN₃. Thin elongated plates measuring ~300 × 100 × 20 μm were obtained after 5 days at room temperature by mixing 1 μl of protein with 1 μl of precipitant solution containing 16% PEG 8K, 0.1 M Mg acetate, and 0.1 M glycine pH 4.5. Crystals were flash-frozen by mixing 75% mother liquor (v/v) with 25% saturated sucrose. X-ray diffraction data were recorded at 100 K (λ = 0.9793 Å) at the LRL-CAT 31-ID beamline Advanced Photon Source (APS) in Chicago. Images were processed using Mosflm version (7.1.1)[51] and scaled with SCALA[52]. Initial phases were obtained by molecular replacement with Phaser[53] based on the DQA1*01:02/DQB1*06:02 α-chain and β-chain from PDB file 1UVQ[34]. One strong molecular replacement solution was found with 1 molecule per asymmetric unit each of the α-chain and β-chain. The solution was confirmed by examination of composite omit maps. After one round of rigid body refinement, the hypocretin peptide was built manually and the whole model improved by cycles of manual building and refinement using COOT[54] and PHENIX REFINE[55], respectively. The overall geometry in the final structure is good, with 98.6% of residues in favored regions, 1.4% in allowed regions of the Ramachandran plot and no outliers. Data collection and refinement statistics are reported (Supplementary Table 1). Residues 105–112 of the β-chain are missing in the electron density, likely due to disorder, and were not included in the structural model. Residues 56–68 of the DQ6-$HCRT_{56–69}$ peptide sequence AGNHAAGILTLGK was built into clear electron density in the peptide-binding cleft, but no electron density was observed for arginine 69 and the linker (GGGGSLVPRGSGGGG) tethering the peptide to the N-terminus of the β chain. A monosaccharide of N-acetyl glucosamine was built at asparagine residues 81 and 121 of the α chain and a disaccharide at asparagine 19 of the β chain. One molecule of Tris was modeled into the electron density. Two amino acid side chains in the β chain were disordered (Arg 48 and Glu 59) and were refined with two alternative conformations. Structural biology software used in this project was curated by SGgrid[56]. Structure figures were generated using the program PyMOL[57].

**In silico analysis using NetMHCIIPan.** We used the MHC-II peptide-binding prediction website, NetMHCIIpan 3.2[33], to evaluate potential DQ6-binding core epitopes within the prepro-HCRT sequence. The resultant in silico predictions of binding rank for HCRT-derived peptides at various lengths are reported (Supplementary Data 1b–d). The peptides containing strong predicted core registers

(Supplementary Fig. 1) were analyzed in experimental binding assays. The NetMHCIIpan 3.2 motif viewer displays binding motifs and predicts that DQ6 (DQA1*0102/DQB1*0602) prefers alanine over all other residues at each of the 9 anchor positions of a potential binding peptides, with small residues, such as serine, glycine, and threonine also preferred at most of the positions.

**DQ6-HCRT modeling and structural analysis.** Models for HCRT-derived peptides bound to DQ6 were developed using the DQ6-$HCRT_{56–69}$ structure. HCRT-derived peptides shown to bind to DQ6 by competition binding studies were docked onto the DQ6-$HCRT_{56–69}$ structure using the 9-aa core epitope defined by NetMHCIIpan for alignment. Peptide side chain rotamers and if necessary DQ side chain rotamers were adjusted using Pymol[57] to accommodate the sequence changes without steric clashes; adjustment of peptide or DQ6 main chain conformation was not required.

**Human subjects and peripheral blood samples.** All donors in this study are HLA-DQB1*06:02⁺. Narcoleptic patients with cataplexy met the criteria for International Classification of Sleep Disorders 3 (ICSD3) for T1N[58]. The controls are either unrelated or influenza-vaccinated subjects. Influenza vaccines included Pandemrix (an AS03-adjuvanted 2009 H1N1 influenza vaccine formulation, GSK) or a seasonal trivalent-inactivated influenza vaccine (TIV, Fluzone, NDC 49281-705-55, 2012–2013 formula, Sanofi Pasteur). PBMCs were received from the Stanford Center for Sleep Sciences and Medicine. Written consent was obtained for collection of all PBMC samples under a Stanford Institutional Review Board approved protocol, following the guidelines for human subjects' research under U. S. Department of Health and Human Services human subjects regulations (45 CFR Part 46).

**Tetramer synthesis.** Customized DQ6-peptide tetramers were all synthesized by NIH Tetramer Core Facility at Emory University using monomers that were secreted from a mammalian cell expression system. These recombinant DQ6-peptide monomers including DQ6-$HCRT_{1–13}$, DQ6-$HCRT_{25–37}$, DQ6-$HCRT_{56–69}$, DQ6-$HCRT_{87–100}$, and DQ6-$EBV_{486–499}$ used identical constructs as mentioned above in the S2 expression system. In each tetramer, peptides are covalently tethered to the N-terminus of DQ6β in order to maintain the peptide specificity.

**In vitro culturing of CD4⁺ T cells.** To test for the presence of DQ6-$HCRT_{peptide}$ tetramer⁺/CD4⁺ T cells, a peptide-loaded antigen-presenting cell (APC) line T2DQ6 (fixed to limit APC proliferation) was co-cultured with T cells for DQ6-restricted antigen stimulation. T2DQ6 was constructed by stable transfection of DQ6 (DQA1*01:02/DQB1*06:02) into T2, a class II-deficient TxB hybrid cell[31]. T2DQ6 cells were maintained in IMDM, GlutaMAX supplemented media (Thermo Fisher Scientific) with 10% HI FBS, 1% penicillin/streptomycin (P/S) and 1 mg/ml G418 (to maintain selective pressure on DQ6 transfectants). To load antigen, T2DQ6 cells were pulsed with peptides (i.e., $EBV_{486–500}$, $HCRT_{56–69}$, $HCRT_{87–100}$, synthesized by Genscript) at 1 μM final concentration and incubated for 6 h at 37 °C. After peptide loading, 10 million T2DQ6 cells were washed with phosphate buffered saline (PBS) and then fixed by incubating with 0.025% glutaraldehyde in 2 ml PBS at RT for 30 s. After the addition of another 2 ml PBS, the cells were incubated for another 10 min at RT. Fixed T2DQ6 was washed twice with PBS and once with complete RPMI (RPMI 1640 medium supplemented with 10% HI human AB serum, 2 mM glutamine and 1% PS, Thermo Fisher Scientific) before mixing with CD4⁺ T cells. Human PBMCs frozen in NUNC tubes were thawed quickly at 37 °C and added slowly to 10 ml warm complete RPMI. PBMCs were pelleted and resuspended in cold buffer for CD4⁺ T cell isolation. Cells were isolated from the PBMCs by negative selection, using the CD4⁺ cell isolation kit, according to the manufacturer's instructions (Miltenyi Biotec). Isolated CD4⁺ T cells were resuspended at $1 × 10^6$ cells/ml in warm complete RPMI and rested for at least 1 h before mixing with fixed T2DQ6 cells that were also resuspended at $1 × 10^6$ cells/ml in warm complete RPMI. A mixture of 1:1 volume ratio of CD4⁺ T cells and T2DQ6 cells in the presence of recombinant human IL-7 at final concentration of 2.5 ng/ml was aliquoted onto a 96-well plate and incubated at 37 °C for 6 days. On day 6 and day 9, 100 μl of the spent medium was removed from each well and replaced with fresh complete RPMI containing recombinant IL-7 at 2.5 ng/ml and IL-2 at 40 U/ml final concentrations. On day 12, a second round of antigen stimulation was performed similarly, using fixed T2DQ6 cells loaded with the corresponding peptides.

**Analysis of tetramer⁺/CD4⁺ cells in the in vitro culture.** Sufficient cells from the co-culture were collected, washed with complete RPMI, and resuspended in 5 ml complete RPMI at RT prior to Ficoll gradient separation. Cells above the Ficoll media were washed and resuspended at $10 × 10^6$ cells/ml in complete RPMI for blocking. After 10 min, tetramers were added to a final concentration of 30 μg/ml and staining was performed for 30 min at 37 °C in the dark, followed by another 15 min incubation at RT with the addition of Alexa fluor 488 anti-human CD4 and PerCP-Cy5.5 anti-human CD19 Abs (BioLegend, to separate T cells from the TxB hybrid T2 cells). Cells were then washed twice with chilled PBS + 10% HI FBS and resuspend in 200 μl PBS + 10% HI FBS for flow cytometric analysis. Live/dead dyes such as propidium iodide (PI, Thermo Fisher Scientific) or Via Probe (7-AAD, BD

Biosciences) were added to each sample before acquisition on a flow cytometer. Cytometers included FACSCalibur, LSR II, and FACSAria II (BD Biosciences).

**Single cell index sorting (iFACS) of tetramer$^+$/CD4$^+$ cells.** Frozen PBMCs were received as randomized pairs each composed of one patient sample with one control sample for a blinded study. The Mellins laboratory performed two sets of independent experiments using PBMCs from 12 patient/control pairs of DQ6$^+$ donors. In Set A, cells of control (C) or patient (P) subjects 1–4 were stained with DQ6-HCRT$_{1-13}$ or DQ6-HCRT$_{56-69}$ tetramer. In Set B, cells of C or P 5–12 were stained with DQ6-HCRT$_{1-13}$ or DQ6-HCRT$_{87-100}$ tetramer and selected samples (C7, 8, 11, 12, and P7–12) were stained with HCRT$_{25-37}$ tetramer (Table 1). Paired PBMC samples were thawed and used for CD4 T cell isolation, as described above. Cell viability was maintained by minimizing the exposure of primary CD4$^+$ T cells to temperatures higher than 4 °C. 3–5 million CD4$^+$ T cells were labeled with LIVE/DEAD cell stains (Life Technologies, Thermo Fisher Scientific) in PBS on ice for 30 min, and then washed and incubated at a density of $10 \times 10^6$ cells/ml in complete RPMI with one of the following tetramers: DQ6-HCRT$_{1-13}$, DQ6-HCRT$_{25-37}$, DQ6-HCRT$_{56-69}$, DQ6-HCRT$_{87-100}$ at a final concentration of 50 μg/ml at 37 °C for 15 min. After the addition of anti-CD4 and anti-CD19 Abs, staining was performed on ice for another 3 h. In Set A, anti-CD127 and anti-CD25 Abs (BioLegend), in addition to anti-CD4 and anti-CD19 Abs, were used to evaluate the subsets of tetramer$^+$/CD4$^+$ T cells in the FACS experiment. Cell samples were then washed in PBS + 1% bovine serum albumin (BSA) and applied on a FACSARIA II cell sorter in the Stanford Shared FACS Facility for the single cell fluorescence-activated cell sorting (FACS). Up to 96 tetramer$^+$/CD4$^+$ cells per sample were individually sorted into a 96-well PCR plate (Eppendorf) with each well containing 10 μl of 1x OneStep RT-PCR buffer (QIAGEN). The index feature associated with the single cell FACS (iFACS) allowed recording of fluorescence intensity (FI) parameters of each sorted cell.

**Index analysis.** Data including index FI values at each channel for sorted single cells were exported from the FACSDIVA software. To determine the tetramer-binding rank of sorted clones within a specific subject-tetramer category, index FI at the tetramer channel normalized by forward scatter (FSC) intensity of each single tetramer$^+$/CD4$^+$ cell was compared with MFI of the tetramer$^{neg}$/CD4$^+$ cell population that was normalized by mean FSC intensity.

**Sequencing of TCR and phenotypic transcripts in single cells.** Single cell mRNA sequencing was performed after three rounds of nested PCR amplification of TCR and phenotypic transcripts using a well-established pipeline[23] with some optimization. Briefly, OneStep RT-PCR (following QIAGEN manual) using single cells as the template in the same 96-well PCR plate into which tetramer$^+$/CD4$^+$ cells were sorted was initiated on the same day when iFACS was accomplished. The annealing temperature was set to 58 °C (used for all three rounds of PCR reactions). This first round of 15 μl multiplex PCR amplified 240–300 base pairs (bps) mRNA sequences of target TCR and phenotypic transcripts by a set of primers recognizing 38 TCRα genes, 36 TCRβ genes, and 25 selected phenotyping marker genes. TCR amplicons cover the V(D)J regions including CDR3 sequences. Two slightly different pairs of specific primers were applied to amplify IFN-γ transcripts in Set A versus Set B donor samples. TCR and phenotypic amplicons from the same cell were then further amplified in separate 96-well PCR plates in a second round of multiplex PCR (15 μl) using 1 μl of the RT-PCR products as the template and HotStarTaq enzyme (QIAGEN) as the DNA polymerase. The second round PCR amplicons (200–250 bps) in selected wells were validated by gel electrophoresis. In the third round of amplification, 1 μl aliquot of the second PCR products (TCR or phenotyping, separately) was used as a template in 15 μl PCR reaction, which incorporates Illumina paired-end (PE) sequences and a unique pair of barcodes with amplicons in each well. The third round PCR amplicons with a length of 350–380 bps from each well were pooled at equal proportion by volume and purified from 2% agarose gel using Qiaquick gel extraction kit (QIAGEN). The incorporated PE sequences enabled deep sequencing on the Illumina Miseq platform (Human Immune Monitoring Center at Stanford University), whereas barcodes allowed deconvolution of deep sequencing data.

**Sequencing data analysis.** The VDJFasta algorithm[23] was used to de-multiplex raw sequencing data and assign each sequence read to a particular well in each PCR plate according to unique plate-row–column barcodes. The average read number per well was 6091 ± 4556. Reads with at least 95% sequence homology were assumed to derive from a consensus sequence of the same TCR. A consensus TCRβ sequence with over 80% reads in a well (BetaConfi >80%) was assigned to the cell. The top one consensus TCRα sequence with over 30% reads in a well (AlphaConfi >30%) was assigned to the cell as the dominant TCRα; whereas a second consensus TCRα sequence with over 10% reads in the same well (altalAphaConfi >10%), if any, was assigned to the cell as the alternative TCRα. For phenotyping markers of the cell, the total reads containing at least 95% sequence homology to a transcription factor or cytokine gene were scored. Both Illumina MiSeq deep sequencing and data analysis were performed at the Human Immune Monitoring Center at Stanford University.

**GLIPH analysis.** The GLIPH algorithm (https://github.com/immunoengineer/gliph) was used to cluster TCRs with a high probability of sharing antigen-binding specificity due to the similarity among their CDR3 sequences[24]. Based on the extent of sequence similarity, three types of conserved motifs were classified: global, local, or single motifs, respectively (Supplementary Data 4a, c). Members in a motif group with significant enrichment of common V genes and clonal expansion were summarized (Supplementary Data 4b, d) to reveal TCR clonotypes isolated from more than one DQ6$^+$ donor or in more than one tetramer categories but from the same donor.

**Construction of Jurkat cell lines expressing candidate TCRs.** To encode the entire α/β chains of a candidate TCR, the nucleotide sequences for CDR3α and CDR3β (Supplementary Data 3) were incorporated in frame with the corresponding V(D)J genes (IMGT/V-QUEST[43]). TCRα/β genes were then synthesized and cloned into a plasmid pEF1a-TCRA_2A_TCRB_IRES-AcGFP1 (by GenScript). Each reconstructed plasmid uses a mammalian promoter, EF1a to direct co-expression of α and β chains of one candidate TCR. TCRA and TCRB are separated by the 2A self-cleaving peptide sequence. For selection purposes, the plasmid also encodes the green fluorescence protein (GFP) and neomycin resistance gene, with expression driven by separate promoters. TCR-encoding plasmids were then transfected into the TCRα/β-deficient Jurkat cell line (J76-NFATRE-luc)[24] by nucleofection following the Amaxa Optimized protocol (Lonza). Transfectants were selected by G418 at a concentration of 1 mg/ml and recovered after 3–4 weeks of culturing in the RPMI medium supplemented with 10% HI FBS, 2 mM glutamine, and 1% PS. To avoid the heterogeneity in TCR expression observed in a polyclonal cell line, TCRαβ$^+$/GFP$^+$/CD3$^+$ cell transfectants were individually sorted by single cell FACS, to expand clonal cell lines originating from single cell transfectants. These clonal lines were then co-stained with PE anti-TCRα/β Abs (BD Biosciences) and APC DQ6-HCRT tetramers to confirm the expression of TCRs and validate their binding to tetramers using flow cytometric analysis.

**T cell activation assay using a luciferase reporter.** The TCR transfectants of J76-NFATRE-luc cells expresses the NFAT-RE (response element)-luciferase reporter gene allowing the conversion of T cell activation signaling to luciferase activity[24]. $1 \times 10^5$ K562-DQ6 cells (an artificial APC line expressing the only HLA allele: DQA1*01:02/DQB1*06:02)[24] and $1 \times 10^5$ single clonal expanded TCR transfectants of J76-NFATRE-luc cells were mixed in 100 μl RPMI medium supplemented with 10% HI FBS, 2 mM glutamine, and 1% PS. The co-culture was incubated with various stimuli at 37 °C for 1 day before quantification of the luciferase activity. Stimuli included 10–50 mM of one HCRT peptide (i.e., HCRT$_{1-13}$, HCRT$_{25-37}$, HCRT$_{87-100}$, and HCRT$_{87-97}$-NH2, synthesized by GenScript), or 1 μg/ml anti-CD3 + 1 μg/ml anti-CD28 Abs (BioLegend) as the positive control, or an equal volume of PBS as the negative control. After incubation, 50 μl of co-culture was mixed with 50 μl of luciferase substrate provided in the Nano-Glo Luciferase Assay System (Promega). The mixture was then transferred to a flat-bottom white 96-well plate (Costar) for a measurement of chemiluminescence using a plate reader.

**Phenotypic analysis.** The number of sequencing reads of a phenotyping marker in a well is dependent on the number of amplicons resulting from three rounds of nested PCR reactions. This number does not necessarily reflect the transcriptional level of the marker per cell, as PCR may bias for transcripts whose amplification occurs relatively efficient at the conditions described above. To eliminate any PCR bias in this semi-quantitative sequencing approach, we assigned 1 or 0 to indicate the presence or absence of reads of a phenotypic transcript in a well without weighing its actual reads. Comparison between patient and control samples was then performed for the overall dataset or TCR clones clustered according to different schemes (Supplementary Data 5b). Numbers of cells from different donors with various tetramer-binding specificities that express each phenotypic transcript were also compared (Supplementary Data 5c).

**Statistical analysis.** A two-tailed $t$-test was used to compare a single variable between two groups of samples. Welch's $t$-test (unequal variance $t$-test) was used when the two groups of samples had unequal variances or unequal sample sizes. The paired samples $t$-test was used when there were correlated pairs of samples in the two groups or experimental pairs show strong correlation on a scatter plot. Mann–Whitney $U$-test was used to compare two groups of non-normally distributed samples. In the luciferase reporter assay, the $t$-test was applied between the tested TCR and the control TCR. $P < 0.05$ was considered as statistical significance. The single cell pipeline and the corresponding analysis tool including GLIPH have high sensitivity with very low false-positive rate[23,24]. Chi-squared test was used to determine the statistical significance of skewing of expanded clonotype detection between two groups. Chi-squared test was also used to determine the statistical significance of skewing of phenotypic parameters within a TCR cluster versus another. Each stack of $P$ values for a set of chi-squared test was further adjusted using the Benjamini, Krieger, and Yekutieli two-stage linear step-up procedure with the desired false discovery rate (FDR) $Q = 5\%$. The null hypothesis that there is no skewing of expanded clonotype detection or there is no skewing of phenotypic parameters between two groups was rejected, only if the $P$ value is less than the

adjusted cut-of value (this is shown in the table of each FDR-controlling procedure). All statistics were performed with GraphPad Prism, using the built-in analysis tool.

**Reporting summary**. Further information on research design is available in the Nature Research Reporting Summary linked to this article.

## Data availability

X-ray structural data for DQ6-HCRT$_{56-69}$ crystallization has been deposited to worldwide protein data bank (https://www.rcsb.org), PDBID: 6DIG; and the structure has been validated. Raw single-cell sequencing data has been deposited to NCBI GEO database (GSE135852). Processed sequencing data are provided in Supplementary Data 2. The source data underlying the figures of this manuscript are provided as a Source Data file. All other relevant data are available from the authors.

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

## Acknowledgements
We thank Eli Lilly Company that operates the Lilly Research Laboratories Collaborative Access Team (LRL-CAT) beamline at Sector 31 of the Advanced Photon Source, which is a U.S. Department of Energy (DOE) Office of Science User Facility operated for the DOE Office of Science by Argonne National Laboratory under Contract No. DE-AC02-06CH11357. We also thank Z. Maben for assistance with crystallization; the NIH Tetramer Facility for providing recombinant HLA-DQ6 tetramers; A. Han and J. Glanville from the Davis Laboratory for helping with the single cell sequencing pipeline and sharing resource code; X. Ji at the Stanford Human Immune Monitoring Center for Miseq sequencing support. This work was funded by GlaxoSmithKline Biologicals SA (GSK), NIAID/NIH (AI-038996), the Child Health Research Institute, Lucile Packard Foundation for Children's Health, as well as the Stanford CTSA (UL1 TR000093). We especially thank R. Van Der Most and S. Buonocore from GSK for the consistent scientific input and feedback from the first data availability.

## Author contributions
W.J. and E.D.M. conceived the project and designed the experiments; W.J., J.R.B., S.H., L.J.S., and E.D.M. analyzed the results. W.J. and S.S. performed the peptide-HLA-binding studies with assistance from L.J.S. and E.D.M.; W.J., and J.R.B. performed crystallization and structural analysis with assistance from G.W., L.L., L.J.S., and E.D.M.; W.J. and C.M. performed tetramer staining studies with assistance from A.I. They and B.K. validated the low frequency of tetramer+/CD4+T cells in the circulation; W.J. performed the single cell sorting and sequencing with assistance from L.A., S.S., and S.A.; W.J., W.W., and S.C. analyzed the sequencing data with assistance from M.M.D., L.T., and E.D.M.; W.J. and S.H. performed TCR validation studies with assistance from H.H.; W.J. and E.D.M. wrote the manuscript with significant input from J.R.B., L.J.S. All authors agreed with the submission.

## Competing interests
The authors declare no competing interests.
