## [Peer Review File · Nature Communications]

Editorial Note: This manuscript has been previously reviewed at another journal that is not operating a transparent peer review scheme. This document only contains reviewer comments and rebuttal letters for versions considered at Nature Communications .

Reviewers' comments:

Reviewer #1 (Remarks to the Author):

In this manuscript Jiang et al. identify and analyze hypocretin-specific, CD4 T cells in narcolepsy type 1, a rare sleep disorder due to loss of neurons producing hypocretin, and in HLA-matched healthy controls. Given its very tight association with the HLA-DQB1*:0602 allele, narcolepsy has long been suspected to be an immune-mediated or even autoimmune disease. This hypothesis has been strengthened more recently by the discovery of TCR alpha and beta loci polymorphisms associated with the disease, and by converging epidemiological studies revealing increased risk of developing narcolepsy following vaccination against the 2009 pH1N1 influenza virus. Moreover, two recent papers identified increased frequency of hypocretin-specific CD4 T cells in people with narcolepsy as compared to HLA-matched healthy controls

Here, the authors characterize with great depth CD4 T cells recognizing HLA-DQ6: hypocretin peptide complexes in both patients and controls. They first identified a series of hypocretin peptides binding to DQ6 and characterized the structure of several HLA-DQ6: hypocretin peptide complexes. Then the authors used 4 different HLA-DQ6: hypocretin tetramers to enumerate and study tetramer+ CD4 T cells in 12 patients and 12 controls. Notably, the authors sequenced TCR and performed a selected transcriptomic single cell analysis on sorted tetramer+ CD4 T cells. They characterized the TCR alpha and beta sequence and the phenotype of in vivo expanded tetramer+ CD4 T cell clones. Finally, they expressed TCR α/β pairs from at least 3 of the tetramer+ clones into TCR-deficient Jurkat cells and could confirm specificity for HLA-DQ6: hypocretin for one of them.

This reviewer acknowledges that the study evaluated with depth HLA-DQ6: hypocretin peptide:TCR interactions that could be the basis for a potentially pathogenic autoreactive response in patients with narcolepsy type 1. However, the results presented in the study mostly underline the absence of significant differences in the frequency and relative affinity of the TCR of the studied tetramer+ CD4 T cells between patients and controls. Moreover, within the limits of this modestly powered study, TCR sequences and transcriptomic profiles could not distinguish the tetramer+ CD4 T cells between patients and controls. In particular, TCR α/β sequencing failed to identify CDR3 motifs that would be statistically associated to narcolepsy. The expanded clones, including TCR27, used public CDR3 sequences in both patients and controls. Furthermore, no indication for stronger in vivo clonal expansion was detected in patients as compared to the paired controls, hence not supporting the pathogenic relevance of the studied tetramer+ CD4 T cells, or their link with disease.

The methodology is of high standard, the results clear but statistical analyses should back all claims of difference between patients and controls, and the authors should refrain from over-interpreting their data.

Specific comments:

In this regard the statement 'In vivo expansion of HCRT-reactive TRAJ24-G allele+ cells provides critical evidence for an autoimmune contribution to narcolepsy development' in the abstract is puzzling given that the frequency of patients and controls exhibiting in vivo expansion of HCRT-reactive TRAJ24-G allele+ CD4 T cells is not statistically different.

This reviewer does not find 'striking' that 2 of 12 patients vs. 0 of 12 controls show in vivo expansion of TCR $\alpha\beta$ clonotypes carrying the L variant. The authors should tone down this statement or show statistical differences between groups.

In the same vein, definite claims such as 'our discovery of eTRAJ24L clonotypes provides novel biomarkers of disease risk/diagnosis' are not warranted

It would be valuable to assess whether the phenotypic features of tetramer+/CD4+ cells expressing the TRAJ24 gene segment differ significantly between patients and controls.

The following comment 'This differs from previous observation of negligible DQ6-HCRT56-69 or DQ6-HCRT87-100 tetramer+ cells from DQ6+ donors' is subjective and inflammatory. It does not belong here.

Fig. 3c Enrichment of DQ6-HCRT56-69 tetramer+/CD4+ T cells in HCRT87-100 peptide stimulated culture: enrichment cannot be assessed, as pre-culture values of the proportion of DQ6-HCRT56-69 tetramer+/CD4+ T cells are not provided.

The following sentences are misleading as they implicitly convey the message that the authors have identified significant differences between patients and controls, whereas they haven't: Therefore, HCRT25-37 lacks a T1N-associated feature. In contrast, DQ6-HCRT87-100 tetramers identified expanded clonotypes in 5/8 cases (highly expanded clonotypes with ≥ 5 isolates seen in 3 cases: P7, P9, and P12) vs 2/8 controls (no highly expanded clonotypes). Similar case/control differences were not observed in the expanded clonotypes isolated by other tetramers.

Having expressed TCR α/β pairs from tetramer+ clones into TCR-deficient Jurkat cells, the authors should test for their cross-reactivity to H1N1 antigens.

Proper statistical analysis should be performed to back the statement 'this finding strengthens our argument that the epitope in HCRT87-100 may be T1N-associated, as none of the 4/8 cases (P7-9 and P12) harboring DQ6-HCRT87-100 tetramer+ multi-tetramer-binding clonotypes was previously H1N1 influenza-vaccinated, whereas the only control C11 harboring multi-tetramer-binding clonotypes identified by the DQ6-HCRT87-100 tetramer was H1N1-vaccinated'.

'HLA class II expression in the central nervous system (CNS) is restricted to microglia and certain neural progenitors'. This is not accurate see for instance Mrdjen et al. Immunity 2018.

'the presentation of ... epitopes to naïve CD4+ T cells may occur through presentation by neuron-associated microglia expressing DQ6': experimental data do not suggest such as scenario (Mundt et al. Science Immunology 2019).

Reviewer #2 (Remarks to the Author):

The authors have clarified many points of confusion from the original review, but these clarifications have raised new concerns.

1) The statistics in Figure 4 are totally inappropriate. The authors claim they have to use this roundabout method for controlling for user variability because "It is impossible to perform cell isolation, single cell sorting, and all three steps of PCR reactions for more than a pair of samples by one person on a single day. In this situation, technical variation becomes as important as biological variation when performing a comparison of all patient samples vs all control samples that are processed on different day as pairs." I'm perplexed by this--most experimental procedures are now multi-day affairs and they don't require this introduction of an artificial technical control. What kind of variation could be introduced that would alter TCR sequences from one day to the next? Flow cytometers are actually quite tractably controlled for variation as well using proper settings and QC management. Consider the vast amounts of sequencing data compared in tumor studies or other immunological studies that are done across months. In the absence of some further clarification, the need for this correction remains unjustified.

But actually more importantly, even if it could be justified, this is an inappropriate analysis. The solution to experimental variation is not to artificially introduce irrelevant comparisons and fail to actually measure the contribution of the experimental variation by using a paired t-test. Building a statistical model (various approaches here would be appropriate) to include day of analysis, technician etc. would allow the actual experimental variation to be weighted. This of course can't be done in Prism, but these data are already too complicated for a Prism analysis. The authors should consult a statistician to assist with this analysis.

2) The authors state that they can't analyze the alpha chain for motifs because of the dual alpha problem, and the overall lower recovery, when they repeatedly do so (granted at a low resolution) throughout the manuscript. Further they state that alpha motif analyzers don't exist (that they are aware of) and that they will wait for their own efforts to develop one. Several are currently available and have been for years, including a very easy to use component of vjtools, and another in TCRdist (which explicitly analyzes paired data). So much of the manuscript is focused on the alpha chain features, it's inexplicable why they would not want to do a more in-depth analysis of these sequences.

Identifying both alpha and beta motifs would strengthen the overall analysis since, as the authors rightly note, truly public paired sequences are very rare, but public paired motifs are exceedingly common in epitope-specific responses.

3. Regarding data deposition and availability--I found no data availability statement as mandated by the checklist, and the statement that they will "look into" the sequencing read archive or other database is not acceptable. The processed sequencing data might be in supplemental material, but the standard in the field is for the raw sequencing reads to be deposited (even if locked) prior to review.

Reviewer #3 (Remarks to the Author):

This is a very big effort by Mellins and collaborators to investigate the finding of CD4 reactive T cells in patients with narcolepsy. Normal patients have been shown to have circulating CD4 T cells to hypocretin peptides, the putative target of this autoimmunity, the response of which is linked to HLA-DQ6.

Mellins et focus on the differences among these T cells in control patients and those found in patients with narcolepsy [T1N]- they go into a series of experimental manipulations and steps to prove clonal expansion on T cells of selected patients w T1N having some of the susceptible gene alleles, proving to their satisfaction that this expansion is a reflection of autoimmune reactivity. Their approach makes sense and is required to prove that the autoreactive process is on.

The work that has gone into this study is notable and shows a huge effort involving different and demanding technical approaches: peptides screening, tetramers, crystallography and then clonal analysis—each step is complex and not easy to judge.

The identification of the epitopes and the structural analysis is excellent and very convincing

However, some of the results are of borderline significance although in toto the whole trajectory is logical—their conclusions albeit on limited number of patients needs to be noted.

I am confused on the display of Table 4c and what the numbers really mean- at the end an issue is that some T cells from the controls do expand ie the point made of 2/8 vs 87-100. And evident in 4a. So although the main point on the genetics for the T1N patients is understood the issue remains that there is expansion in some controls, regardless of its correlation with genetic markers. Concerning the

discussion on this issue related to controls and vaccination, are not the numbers too limited to make any conclusions? This point needs better analysis/discussion.

The amounts of work in the experiment in 6e is major yet the differences noted are small- the data shown in 6e is not convincing. This section needs further evaluation.

Reviewer #1 (Remarks to the Author):

*In this manuscript Jiang et al. identify and analyze hypocretin-specific, CD4 T cells in narcolepsy type 1, a rare sleep disorder due to loss of neurons producing hypocretin, and in HLA-matched healthy controls. Given its very tight association with the HLA-DQB1*0602 allele, narcolepsy has long been suspected to be an immune-mediated or even autoimmune disease. This hypothesis has been strengthened more recently by the discovery of TCR alpha and beta loci polymorphisms associated with the disease, and by converging epidemiological studies revealing increased risk of developing narcolepsy following vaccination against the 2009 pH1N1 influenza virus. Moreover, two recent papers identified increased frequency of hypocretin-specific CD4 T cells in people with narcolepsy as compared to HLA-matched healthy controls.*

Here, the authors characterize with great depth CD4 T cells recognizing HLA-DQ6: hypocretin peptide complexes in both patients and controls. They first identified a series of hypocretin peptides binding to DQ6 and characterized the structure of several HLA-DQ6: hypocretin peptide complexes. Then the authors used 4 different HLA-DQ6: hypocretin tetramers to enumerate and study tetramer+ CD4 T cells in 12 patients and 12 controls. Notably, the authors sequenced TCR and performed a selected transcriptomic single cell analysis on sorted tetramer+ CD4 T cells. They characterized the TCR alpha and beta sequence and the phenotype of in vivo expanded tetramer+ CD4 T cell clones. Finally, they expressed TCR α/β pairs from at least 3 of the tetramer+ clones into TCR-deficient Jurkat cells and could confirm specificity for HLA-DQ6: hypocretin for one of them.

This reviewer acknowledges that the study evaluated with depth HLA-DQ6: hypocretin peptide:TCR interactions that could be the basis for a potentially pathogenic autoreactive response in patients with narcolepsy type 1. However, the results presented in the study mostly underline the absence of significant differences in the frequency and relative affinity of the TCR of the studied tetramer+ CD4 T cells between patients and controls. Moreover, within the limits of this modestly powered study, TCR sequences and transcriptomic profiles could not distinguish the tetramer+ CD4 T cells between patients and controls. In particular, TCR α/β sequencing failed to identify CDR3 motifs that would be statistically associated to narcolepsy. The expanded clones, including TCR27, used public CDR3 sequences in both patients and controls. Furthermore, no indication for stronger in vivo clonal expansion was detected in patients as compared to the paired controls, hence not supporting the pathogenic relevance of the studied tetramer+ CD4 T cells, or their link with disease.

The methodology is of high standard, the results clear but statistical analyses should back all claims of difference between patients and controls, and the authors should refrain from over-interpreting their data.

Specific comments:

In this regard the statement 'In vivo expansion of HCRT-reactive TRAJ24-G allele+ cells provides critical evidence for an autoimmune contribution to narcolepsy development' in the abstract is puzzling given that the frequency of patients and controls exhibiting in vivo expansion of HCRT-reactive TRAJ24-G allele+ CD4 T cells is not statistically different.

Response:

We have toned down our statement in the revision. We include the idea that a demonstration of HCRT-reactive (signaling), TRAJ24-G TCR+ cells that are clonally expanded in T1N patients opens an avenue for further investigation of autoimmunity in narcolepsy.

This reviewer does not find 'striking' that 2 of 12 patients vs. 0 of 12 controls show in vivo expansion of TCR α/β clonotypes carrying the L variant. The authors should tone down this statement or show statistical differences between groups.

Response:

We have removed the word 'striking' and toned down all related statements in the revision. We have also provided additional statistical analyses comparing patients and controls in Fig. 4c and Supplementary Fig. 5 for TCRs and in Supplementary Data 6 for phenotypes.

In the same vein, definite claims such as 'our discovery of eTRAJ24L clonotypes provides novel biomarkers of disease risk/diagnosis' are not warranted.

Response: We have edited the text in the revision.

It would be valuable to assess whether the phenotypic features of tetramer+/CD4+ cells expressing the TRAJ24 gene segment differ significantly between patients and controls.

Response: Please refer to Supplementary Data 6 for details. Briefly, tetramer+/CD4+/TRAJ24+ cells in patients express PRF1 (P=0.0105) and TGF- β (P=0.0016) significantly more frequently compared to cells in controls, but express IFN γ (P=0.0007) significantly less frequently. In addition, we have edited the text in the revision.

The following comment 'This differs from previous observation of negligible DQ6-HCRT56-69 or DQ6-HCRT87-100 tetramer+ cells from DQ6+ donors' is subjective and inflammatory. It does not belong here.

Response: We have removed the comment in the revision.

Fig. 3c Enrichment of DQ6-HCRT56-69 tetramer+/CD4+ T cells in HCRT87-100 peptide stimulated culture: enrichment cannot be assessed, as pre-culture values of the proportion of DQ6-HCRT56-69 tetramer+/CD4+ T cells are not provided.

Response: To allow assessment of enrichment, we have modified Fig. 3c to include control data at "no peptide" condition. We removed the unnecessary control data from the "HCR1-13 peptide" condition.

The following sentences are misleading as they implicitly convey the message that the authors have identified significant differences between patients and controls, whereas they haven't: Therefore, HCRT25-37 lacks a T1N-associated feature. In contrast, DQ6-HCRT87-100 tetramers identified expanded clonotypes in 5/8 cases (highly expanded clonotypes with ≥ 5 isolates seen in 3 cases: P7, P9, and P12) vs 2/8 controls (no highly expanded clonotypes). Similar case/control differences were not observed in the expanded clonotypes isolated by other tetramers.

Response: We have toned down all related statements in the revision and performed additional statistical analyses in Fig. 4c. The chi-squared test (P=0.0003) shows that there is a significantly higher likelihood of detecting expanded DQ6-HCRT87-100 tetramer+ cells in patients compared to controls (Fig. 4c). This effect on cells isolated with DQ6-HCRT87-100 tetramer differs from the other three DQ6-HCRTpeptide tetramer specificities that yielded a skewed detection of more expanded clonotypes in control samples

Having expressed TCR α/β pairs from tetramer+ clones into TCR-deficient Jurkat cells, the authors should test for their cross-reactivity to H1N1 antigens.

Response: Given the reported T1N association with the 2009 H1N1 pandemic, we agree that TCR cross-reactivity to H1N1 antigens is an important area of investigation. However, it is beyond the scope of this report. In addition, for peptide presentation to Jurkat transfectants, we used APC (K562-DQ6) that do not have the intracellular antigen processing machinery needed for presenting protein antigens (e.g., from H1N1). [REDACTED]. We will address this important question in further work.

[REDACTED]

Proper statistical analysis should be performed to back the statement 'this finding strengthens our argument that the epitope in HCRT87-100 may be T1N-associated, as none of the 4/8 cases (P7-9 and P12) harboring DQ6-HCRT87-100 tetramer+ multi-tetramer-binding clonotypes was previously H1N1 influenza-vaccinated, whereas the only control C11 harboring multi-tetramer-binding clonotypes identified by the DQ6-HCRT87-100 tetramer was H1N1-vaccinated'.

Response:

We have edited related statements in the revision and performed additional statistical analyses in Fig. 4c. We also find an effect of excluding recently TIV-vaccinated cases and controls when assessing case/control differences. Indeed, the DQ6-HCRT87-100 tetramer detected 10 expanded clonotypes (24 isolates) in 4/7 cases (excluding TIV-vaccinated P5), but none in 5/5 controls (excluding TIV-vaccinated C5, C6, C11), with $P < 0.0001$ in a chi-squared test assessing the frequency of expanded cells between patient and control cohorts (**Fig. 4a, c**). The fact that none of the 4 patient subjects was previously influenza-vaccinated suggests the epitope in HCRT87-100 is relevant to *the in vivo* expansion of identified clones in these T1N patients.

'HLA class II expression in the central nervous system (CNS) is restricted to microglia and certain neural progenitors'. This is not accurate see for instance Mrdjen et al. Immunity 2018.

Response:

We have edited our discussion in the revision.

'the presentation of ... epitopes to naïve CD4+ T cells may occur through presentation by neuron-associated microglia expressing DQ6': experimental data do not suggest such as scenario (Mundt et al. Science Immunology 2019).

Response:

We have edited our discussion in the revision.

Reviewer #2 (Remarks to the Author):

The authors have clarified many points of confusion from the original review, but these clarifications have raised new concerns.

1) The statistics in Figure 4 are totally inappropriate. The authors claim they have to use this roundabout method for controlling for user variability because "It is impossible to perform cell isolation, single cell sorting, and all three steps of PCR reactions for more than a pair of samples

by one person on a single day. In this situation, technical variation becomes as important as biological variation when performing a comparison of all patient samples vs all control samples that are processed on different day as pairs." I'm perplexed by this--most experimental procedures are now multi-day affairs and they don't require this introduction of an artificial technical control. What kind of variation could be introduced that would alter TCR sequences from one day to the next? Flow cytometers are actually quite tractably controlled for variation as well using proper settings and QC management. Consider the vast amounts of sequencing data compared in tumor studies or other immunological studies that are done across months. In the absence of some further clarification, the need for this correction remains unjustified. But actually more importantly, even if it could be justified, this is an inappropriate analysis. The solution to experimental variation is not to artificially introduce irrelevant comparisons and fail to actually measure the contribution of the experimental variation by using a paired t-test. Building a statistical model (various approaches here would be appropriate) to include day of analysis, technician etc. would allow the actual experimental variation to be weighted. This of course can't be done in Prism, but these data are already too complicated for a Prism analysis. The authors should consult a statistician to assist with this analysis.

Response:

As requested, we have reported the P-values from unpaired t-test (Fig. 4b: P=0.9741 for HCRT1-13, P=0.4335 for HCRT25-37, P=0.7331 for HCRT56-69, P=0.0110 for HCRT87-100) in the revised manuscript. In addition, we have consulted Professor Lu Tian, PhD (added as an author), and other biostatisticians from Stanford. They support using the paired t-test in this situation, and at their suggestion, we report the result in the supplementary data. As shown in the new Supplementary Fig. 5a, there is a strong correlation between controls and patients within the same pair, i.e., from the same experiment. From a statistical point of view, this provides the rationale for the application of paired t-test.

2) The authors state that they can't analyze the alpha chain for motifs because of the dual alpha problem, and the overall lower recovery, when they repeatedly do so (granted at a low resolution) throughout the manuscript. Further they state that alpha motif analyzers don't exist (that they are aware of) and that they will wait for their own efforts to develop one. Several are currently available and have been for years, including a very easy to use component of vjtools, and another in TCRdist (which explicitly analyzes paired data). So much of the manuscript is focused on the alpha chain features, it's inexplicable why they would not want to do a more in-depth analysis of these sequences.

Identifying both alpha and beta motifs would strengthen the overall analysis since, as the authors rightly note, truly public paired sequences are very rare, but public paired motifs are exceedingly common in epitope-specific responses.

Response:

We agree with the value of motif analysis of alpha chain sequences and apologize for our previous response. Our original goal was to use GLIPH to pull public features out of our dataset, as this type of analysis identifies GLIPH-defined conserved regions used by different but related CDR3 sequences. Beta chain analysis was sufficient for us to prove the only significant public candidate was E%DRGRSET (also see Glanville et al, 2017). In the revised manuscript, we report the GLIPH alpha chain analysis and focus on public features of paired motifs in the new Figure 6. In addition, we have added a paragraph to discuss the results in the revision.

3. Regarding data deposition and availability--I found no data availability statement as mandated by the checklist, and the statement that they will "look into" the sequencing read archive or other database is not acceptable. The processed sequencing data might be in supplemental material, but the standard in the field is for the raw sequencing reads to be deposited (even if locked) prior to review.

Response:

We have updated the “Data Availability” statement with the depository details. Our deposit of raw sequencing files at NCBI GEO database has been approved (GSE135852, will be released in 1 yr). Please use the following token for access to our private records: epqfwqogxzaznyb.

Reviewer #3 (Remarks to the Author):

This is a very big effort by Mellins and collaborators to investigate the finding of CD4 reactive T cells in patients with narcolepsy. Normal patients have been shown to have circulating CD4 T cells to hypocretin peptides, the putative target of this autoimmunity, the response of which is linked to HLA-DQ6.

Mellins et al focus on the differences among these T cells in control patients and those found in patients with narcolepsy [T1N]- they go into a series of experimental manipulations and steps to prove clonal expansion on T cells of selected patients w T1N having some of the susceptible gene alleles, proving to their satisfaction that this expansion is a reflection of autoimmune reactivity. Their approach makes sense and is required to prove that the autoreactive process is on.

The work that has gone into this study is notable and shows a huge effort involving different and demanding technical approaches: peptides screening, tetramers, crystallography and then clonal analysis—each step is complex and not easy to judge. The identification of the epitopes and the structural analysis is excellent and very convincing.

However, some of the results are of borderline significance although in toto the whole trajectory is logical—their conclusions albeit on limited number of patients needs to be noted.

I am confused on the display of Table 4c and what the numbers really mean- at the end an issue is that some T cells from the controls do expand ie the point made of 2/8 vs 87-100. And evident in 4a. So although the main point on the genetics for the T1N patients is understood the issue remains that there is expansion in some controls, regardless of its correlation with genetic markers. Concerning the discussion on this issue related to controls and vaccination, are not the numbers too limited to make any conclusions? This point needs better analysis/discussion.

Response:

The original Supplementary Data 4 provided detailed counts to help readers understand Fig. 4a. However, we realize that it might have been confusing and redundant. To improve clarity, we have modified Figure 4a to include all necessary information and removed Supplementary Data 4 (including Table 4c). In addition, we have added new statistical analysis including a chi-squared test in Fig. 4c and Supplementary Fig. 5 and provided data interpretation and discussion on the issue related to controls and vaccination in the revision. Briefly, the chi-squared test shows that the DQ6-HCRT₈₇₋₁₀₀ tetramer identified significantly more expanded cells in patients compared to in controls. The recent TIV vaccination significantly increased the likelihood of isolating HCRT-tetramer+ clonotypes that bear cross-binding capacity. While this may suggest some relevant biology (i.e., molecular mimicry) for future investigation, it is also informative to perform T1N analysis after excluding TIV-vaccinated subjects, and we report this result.

The amounts of work in the experiment in 6e is major yet the differences noted are small- the data shown in 6e is not convincing. This section needs further evaluation.

Response:

The moderate signal was likely due to the usage of a non-physiological form of epitopes in the HCRT₈₇₋₁₀₀ peptide. In Fig. 7e in the revision, we show a comparison of signaling between TCR27 and the control TCR (P=0.00004) in response to HCRT87-97-NH2. The difference is significantly higher than those using HCRT87-100 (non-amidated form), as shown in Supplementary Fig. 7i (previous Fig. 6f). We have modified the corresponding sections in the revision.

REVIEWERS' COMMENTS:

Reviewer #1 (Remarks to the Author):

This reviewer still fails to be convinced that this study has identified disease-relevant autoreactive T cell features as there was no significant difference in the frequency of DQ6-HCRT tetramers+ cells, in the tetramer fluorescence intensity, and in the frequency of TRAJ24-G allele+ clonotypes between T1N and controls. One notable finding by the authors though is the expression of a TCR α/β pair from tetramer+ clones into TCR-deficient Jurkat cells and the demonstration of its reactivity to the amidated HCRT87-97 peptide. However, given the previous PNAS paper from the Mignot team, the novelty and scientific advance appear quite limited to deserve publication in such a leading Journal.

If expanded DQ6-HCRT87-100 tetramers+ clonotypes are present in 5/8 patients vs. 2/8 controls, as shown in Fig 4A, the 'likelihood of detecting expanded DQ6-HCRT87-100 tetramer+ cells in patients compared to controls' is not 'significantly higher'. May be it is the frequency of the in vivo expanded DQ6-HCRT87-100 tetramers+ clonotypes that differs between T1N and controls. Given the issues with statistical use raised by several reviewers, I think a statistician from the Journal should review this aspect.

Related to the use of statistics, I am still wondering what is the biological significance of the statistical comparisons shown in Fig 3D, E and Supp. Fig. 5B. Whereas the comparison between patients and controls is relevant but does not reveal differences, the comparison between tetramers has much less relevance, only showing trivially that some self-peptides are more recognized as others.

A posteriori exclusion of TIV-vaccinated subjects seems odd to this reviewer. This should have been done either by excluding these individuals from the study or, alternatively, by predefining 2 groups (vaccinated or not) in the study design.

Fig 6C. It is puzzling that CDR3a sequences use a TRBJ gene segment. Please correct the unfortunate typo.

Reviewer #2 (Remarks to the Author):

All concerns have been addressed.

Reviewer #3 (Remarks to the Author):

The authors have made a major effort to answer my critiques and am satisfied with their answers.

Reviewer #1 (Remarks to *the Author*):

This reviewer still fails to be convinced that this study has identified disease-relevant autoreactive T cell features as there was no significant difference in the frequency of DQ6-HCRT tetramers+ cells, in the tetramer fluorescence intensity, and in the frequency of TRAJ24-G allele+ clonotypes between T1N and controls. One notable finding by the authors though is the expression of a TCR α/β pair from tetramer+ clones into TCR-deficient Jurkat cells and the demonstration of its reactivity to the amidated HCRT87-97 peptide. However, given the previous PNAS paper from the Mignot team, the novelty and scientific advance appear quite limited to deserve publication in such a leading Journal.

Response:

Structural analysis of epitopes, demonstration of *in vivo* clonal expansion of TRAJ24-G allele+ TCR α/β pairs in patients, and determination of associated phenotypic features are all contributions from this work. Significantly higher frequencies of *in vivo* expanded DQ6-HCRT87-100 tetramer+ cells are seen in patients (Table 2). The functional reactivity of TRAJ24-G allele+ clonotype to the amidated HCRT87-97 peptide indicates its specificity for a post-translational modification of hypocretin, consistent with an autoimmune hypothesis for narcolepsy. The identification of a public clonotype with this specificity from a direct *ex vivo* screen is novel.

If expanded DQ6-HCRT87-100 tetramers+ clonotypes are present in 5/8 patients vs. 2/8 controls, as shown in Fig 4A, the 'likelihood of detecting expanded DQ6-HCRT87-100 tetramer+ cells in patients compared to controls' is not 'significantly higher'. Maybe it is the frequency of the *in vivo* expanded DQ6-HCRT87-100 tetramers+ clonotypes that differs between T1N and controls. Given the issues with statistical use raised by several reviewers, I think a statistician from the Journal should review this aspect.

Response:

Sorry for the confusion. This comparison is between the frequency of expanded cells detected by DQ6-HCRT87-100 tetramer in cases versus controls (Figure 4C, we have changed this panel into Table 2), rather than number of subjects having expanded cells. We have clarified this in the revision, as follows. DQ6-HCRT87-100 tetramer detected significantly ($P=0.0003$, chi-squared test) more expanded cells in patients (26/740, 3.51%) than in controls (6/751, 0.8%).

Related to the use of statistics, I am still wondering what is the biological significance of the statistical comparisons shown in Fig 3D, E and Supp. Fig. 5B. Whereas the comparison between patients and controls is relevant but does not reveal differences, the comparison between tetramers has much less relevance, only showing trivially that some self-peptides are more recognized as others.

Response:

We have removed the "Set 1" layer of statistical analysis shown in Fig 3D, E and Supp. Fig. 5B. The comparable frequencies of tetramer+ cells (including expanded and unexpanded) in patients and controls provide the basis to compare frequencies of tetramer+ cells that are expanded *in vivo* (e.g., Table 2). If the frequencies were different, the comparison of expanded cells in terms of absolute frequency and relative frequency may not be consistent and hard to interpret.

A posteriori exclusion of TIV-vaccinated subjects seems odd to this reviewer. This should have

been done either by excluding these individuals from the study or, alternatively, by predefining 2 groups (vaccinated or not) in the study design.

Response:

A relationship between vaccination history and narcolepsy development was unclear at the study design stage, and patients with TIV vaccination history were not excluded. However, after analyzing the data, we found that recent TIV vaccination significantly altered the frequency of expanded clones in controls. Therefore, for scientific objectivity, we presented both results including TIV-vaccinated subjects and results excluding these subjects (Table 2).

Fig 6C. It is puzzling that CDR3a sequences use a TRBJ gene segment. Please correct the unfortunate typo.

Response:

We have corrected the typo.